# Room-temperature valence transition in a strain-tuned perovskite oxide

Vipul Chaturvedi [1], Supriya Ghosh [1], Dominique Gautreau[1,2], William M. Postiglione [1], John E. Dewey [1], Patrick Quarterman[3], Purnima P. Balakrishnan[3], Brian J. Kirby [3], Hua Zhou [4], Huikai Cheng[5], Amanda Huon [6], Timothy Charlton [6], Michael R. Fitzsimmons [6,7], Caroline Korostynski[1], Andrew Jacobson[1], Lucca Figari[1], Javier Garcia Barriocanal[8], Turan Birol [1], K. Andre Mkhoyan [1] & Chris Leighton [1] ✉

Cobalt oxides have long been understood to display intriguing phenomena known as spin-state crossovers, where the cobalt ion spin changes vs. temperature, pressure, etc. A very different situation was recently uncovered in praseodymium-containing cobalt oxides, where a first-order coupled spin-state/structural/metal-insulator transition occurs, driven by a remarkable praseodymium valence transition. Such valence transitions, particularly when triggering spin-state and metal-insulator transitions, offer highly appealing functionality, but have thus far been confined to cryogenic temperatures in bulk materials (e.g., 90 K in $Pr_{1-x}Ca_xCoO_3$). Here, we show that in thin films of the complex perovskite $(Pr_{1-y}Y_y)_{1-x}Ca_xCoO_{3-\delta}$, heteroepitaxial strain tuning enables stabilization of valence-driven spin-state/structural/metal-insulator transitions to at least 291 K, i.e., around room temperature. The technological implications of this result are accompanied by fundamental prospects, as complete strain control of the electronic ground state is demonstrated, from ferromagnetic metal under tension to nonmagnetic insulator under compression, thereby exposing a potential novel quantum critical point.

Spin-state crossovers and transitions, wherein the spin-state of an ion gradually or abruptly changes vs. temperature, pressure, or some other stimulus, have long gathered scientific and technological interest. Such crossovers and transitions occur in both inorganic and organic systems, are relevant in contexts as varied as complex oxide magnetism[1–7], metalorganic chemistry[8–10], and earth sciences[11–13], and offer functionality as diverse as switching[9,10], non-volatile memory[9,10], and mechanocaloric refrigeration[14]. Perovskite Co oxides, the archetype being $LaCoO_3$, have been at the center of this activity for

>50 years[1–7]. In $LaCoO_3$, the crystal field splitting of Co $d$ levels by the surrounding octahedral $O^{2-}$ ions barely exceeds the Hund intra-atomic exchange energy, leading to a zero-spin ($S = 0$, $t_{2g}^6 e_g^0$) $Co^{3+}$ ground state and diamagnetism[1–7]. The balance between crystal field splitting, Hund exchange, and Co $3d$ - O $2p$ hybridization is so delicate, however, that by 30 K thermal population of finite spin-states (finite $e_g$ occupancy) already occurs, inducing a gradual spin-state crossover, i.e., thermally-excited paramagnetism[1–7]. The exact nature of the excited spin-states in $LaCoO_3$ with temperature, pressure, doping, and

[1]Department of Chemical Engineering and Materials Science, University of Minnesota, Minneapolis, MN 55455, USA. [2]School of Physics and Astronomy, University of Minnesota, Minneapolis, MN 55455, USA. [3]NIST Center for Neutron Research, National Institute of Standards and Technology, Gaithersburg, MD 60439, USA. [4]Advanced Photon Source, Argonne National Laboratory, Lemont, IL 60439, USA. [5]Thermo Fisher Scientific, Hillsboro, OR 97124, USA. [6]Neutron Scattering Division, Oak Ridge National Lab, Oak Ridge, TN 37830, USA. [7]Department of Physics and Astronomy, University of Tennessee, Knoxville, TN 37996, USA. [8]Characterization Facility, University of Minnesota, Minneapolis, MN 55455, USA. ✉e-mail: leighton@umn.edu

substitution, remain a matter of debate, however, testament to the theoretical and experimental challenge posed by this problem.

More recently, certain doped perovskite cobaltites, specifically those containing Pr, have been understood to add a new dimension to spin-state phenomena. In 2002, Tsubouchi et al. reported the successful bulk synthesis of $Pr_{0.5}Ca_{0.5}CoO_{3-\delta}$, discovering a 90-K spin-state transition, marked by a sharp decrease in magnetic susceptibility on cooling[15]. While a spin-state crossover in such a cobaltite would not be entirely surprising, this was found to be a first-order spin-state transition in $Pr_{0.5}Ca_{0.5}CoO_{3-\delta}$, occurring with simultaneous structural and metal-insulator transitions (from metallic to insulating on cooling); the structural transition also has unusual isomorphic ($Pnma \rightarrow Pnma$) character, involving a large (~2%) contraction of the cell volume[15–20]. The first-order spin-state transition, the simultaneous structural/metal-insulator transitions (which are highly reminiscent of classic metal-insulator systems such as vanadium oxides[1,21]), and the unusual nature of the structural transformation were all new in cobaltites, and starkly different to $LaCoO_3$, generating immediate interest (e.g.,[15–20,22–28]). This activity was further fueled by the discovery that isovalent substitution with the smaller ion $Y^{3+}$, in $(Pr_{1-y}Y_y)_{1-x}Ca_xCoO_{3-\delta}$ (PYCCO), stabilizes the low-temperature collapsed-cell-volume state, driving the spin-state/structural/metal-insulator transition up to ~135 K in $x = 0.30$, $y = 0.15$ bulk materials[18,22–24,27]. Such isovalent substitutions[18,22–24,27,28] thus not only increase the transition temperature, but also stabilize the transition at lower $x$ (lower Co valence), where synthesis challenges related to O vacancy formation are less problematic.

Subsequent exploration established that these phenomena occur only in Pr-containing cobaltites, and that the Pr ions drive the spin-state and metal-insulator transitions through a remarkable mechanism. Specifically, numerous techniques confirm a Pr valence transition, where the typical $Pr^{3+}$ at room temperature abruptly shifts towards $Pr^{4+}$ at low temperature[18–20,23,27,28]. The Co ions in compounds such as $Pr_{0.5}Ca_{0.5}CoO_{3-\delta}$ must then transition from average valence 3.5+ towards 3+, i.e., formally-$Co^{4+}$ ions are converted to $Co^{3+}$. This abruptly decreases the hole density, thereby driving the spin-state and metal-insulator transitions[15–20,22–28]. The unique first-order coupled spin-state/structural/metal-insulator transitions in Pr-based cobaltites are thus valence transitions, occurring at a transition temperature $T_{vt}$.

The central issue then becomes the origin of the Pr valence transition in these materials. In general, anomalous valence behavior in rare-earth (R)-containing compounds has a fascinating history, across diverse materials systems. Beyond the well-known anomalous valence tendencies of Eu[29], it has been known for some time that the entire $RBa_2Cu_3O_{7-x}$ series displays high-temperature superconductivity, with the sole exception of $PrBa_2Cu_3O_{7-x}$, which is insulating and antiferromagnetic[30]. This is thought to arise from a unique ability of Pr $4f$ electrons to hybridize and bond with O $2p$ electrons, forming so-called Fehrenbacher-Rice states near the Fermi energy[31]. In $ATMO_3$ perovskites ($TM$ = transition metal) Goodenough in fact pointed out as early as 2001[32] that Pr is the only rare-earth or lanthanide A-site ion likely to have $4f$ states near the Fermi energy. In the $RCoO_3$ series this has been verified through density functional theory (DFT) calculations[33], and active Pr-O bonding was deduced from anomalous structural behavior in bulk $Pr_{0.5}Sr_{0.5}CoO_{3-\delta}$[34]. The possibility of R valence changes vs. pressure and strain have also been discussed theoretically in $RMnO_3$ systems[35]. In terms of thermal valence transitions, the extraordinary behavior in cobaltites such as PYCCO is potentially analogous to transitions seen in certain Fe-based complex oxides[36], as well as Sm monochalcogenides such as SmS, where a pressure- and temperature-dependent Sm valence transition arises[37]. Various advances have thus been made with the understanding of R valence transitions, but in largely disconnected areas, with little general understanding. In the context of PYCCO, while DFT captures the Pr

valence shift due to $4f$ states traversing the Fermi level as the structure transforms[17], there remains no understanding of what precedes, triggers, or controls the structural transformation at $T_{vt}$.

An important point to emphasize regarding PYCCO and related bulk systems is that the coupled valence/structural/spin-state/metal-insulator transitions are cryogenic phenomena. The $T_{vt} \approx 90$ K in $Pr_{0.5}Ca_{0.5}CoO_{3-\delta}$[15–20] can be increased to ~135 K in $(Pr_{0.85}Y_{0.15})_{0.7}Ca_{0.3}CoO_{3-\delta}$[18,22–24,27], but further Y substitution induces sample quality issues likely associated with solubility limits and/or O vacancy formation[24], frustrating attempts to increase $T_{vt}$. The latter is an important goal, however, as near-ambient control over spin-state transitions, metal-insulator transitions, and particularly valence transitions, would offer highly desirable unrealized device functionality in perovskite oxides. Beyond chemical pressure in bulk materials, heteroepitaxial strain control of thin films is an attractive alternative to tune and enhance properties of perovskite oxides[38,39], illustrative examples being strain-induced ferroelectricity in $SrTiO_3$ (STO)[40,41] and strain control over the electronic/magnetic ground state in rare-earth nickelates[42,43]. In this context, strained epitaxial films of PYCCO and related cobaltites become of high interest, recent work pointing to promise in this approach. Padilla-Pantoja et al.[44], for example, observed a strong influence of the 0.4% tensile strain in $LaAlO_3$/$Pr_{0.5}Ca_{0.5}CoO_{3-\delta}$ films, which induced unusual O vacancy ordering and a ferromagnetic (F) metallic ground state with Curie temperature $T_C \approx$ 170 K, in stark contrast to bulk. This indicates destabilization of the valence transition to a low-temperature insulating state, in favor of a F metal with $T_C$ substantially enhanced over bulk $Pr_{1-x}Ca_xCoO_{3-\delta}$. (Away from $x \approx 0.5$, bulk $Pr_{1-x}Ca_xCoO_{3-\delta}$ exhibits F order, but with maximum $T_C \approx 75$ K[16,45]). No broader study of the influence of tensile and compressive strain using multiple substrates has yet been reported, however, and in fact scattered studies on closer lattice-matched substrates have reported only diffuse, highly-broadened versions of the PYCCO valence/structural/spin-state/metal-insulator transition[46,47].

Here, we provide the first complete study of the influence of heteroepitaxial strain on the structural, electronic transport, and magnetic properties of PYCCO films. Synchrotron X-ray diffraction (SXRD), scanning transmission electron microscopy (STEM) with energy dispersive X-ray spectroscopy (EDS) and electron energy loss spectroscopy (EELS), and atomic force microscopy (AFM) reveal smooth, single-phase, epitaxial films, fully strained to YAlO_3(101) (YAO), $SrLaAlO_4$(001) (SLAO), $LaAlO_3$(001) (LAO), $SrLaGaO_4$(001) (SLGO), and $La_{0.18}Sr_{0.82}Al_{0.59}Ta_{0.41}O_3$(001) (LSAT) substrates, generating biaxial strains between −2.10% (compressive) and +2.34% (tensile). Temperature ($T$)-dependent resistivity ($\rho$), magnetization ($M$), and polarized neutron reflectometry (PNR) measurements reveal complete control over the electronic and magnetic ground state of PYCCO, from F metallic with $T_C \approx 85$ K under large tensile strain, to nonmagnetic and strongly insulating under large compressive strain, a trend that is reproduced by first-principles calculations. YAO/PYCCO films (−2.10% strain) in fact exhibit first-order metal-insulator transitions strain-stabilized to at least 291 K, i.e., around room temperature. $T$-dependent SXRD and EELS measurements confirm simultaneous structural and Co valence/spin-state transitions, establishing a strikingly similar first-order coupled structural/valence/spin-state/metal-insulator transition to bulk, simply enhanced to ambient temperatures. We additionally verify the same qualitative behavior in $Pr_{1-x}Ca_xCoO_{3-\delta}$, demonstrating that epitaxial strain tuning can achieve such control even in the absence of Y chemical pressure. Finally, we create "strain phase diagrams" for both $(Pr_{0.85}Y_{0.15})_{0.7}Ca_{0.3}CoO_{3-\delta}$ and $Pr_{0.7}Ca_{0.3}CoO_{3-\delta}$ films, highlighting the possibility of exploring a novel spin-state quantum critical point in these materials.

## Results
Shown first in Fig. 1a-f are specular SXRD scans around the 002 reflections of ~30-unit-cell-thick PYCCO films grown on YAO, SLAO,

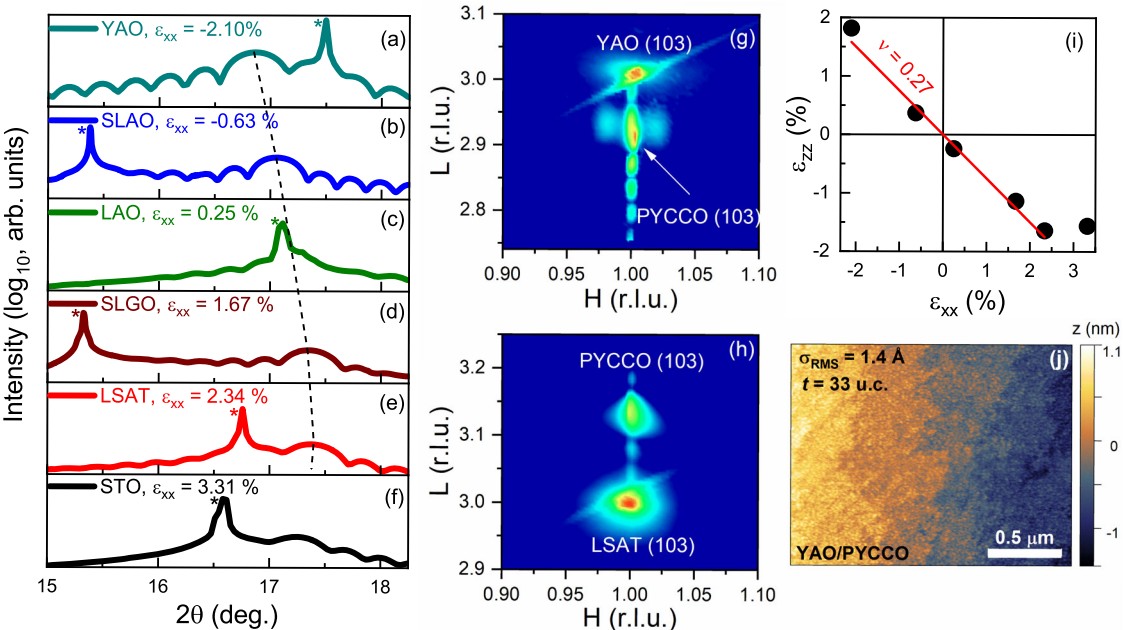

**Fig. 1 | Structural characterization of $(Pr_{0.85}Y_{0.15})_{0.7}Ca_{0.3}CoO_{3-\delta}$ films.** Specular synchrotron X-ray diffraction (SXRD, 0.564 Å wavelength) around the 002 film peaks of 28- to 33-unit-cell-thick $(Pr_{0.85}Y_{0.15})_{0.7}Ca_{0.3}CoO_{3-\delta}$ films on (**a**) YAO(101), (**b**) SLAO(001), (**c**) LAO(001), (**d**) SLGO(001), (**e**) LSAT(001) and (**f**) STO(001) substrates (labeled with the "in-plane strain", $\varepsilon_{xx}$). Substrate reflections are labeled "*". Asymmetric SXRD reciprocal space maps around the film 103 peaks (pseudocubic notation) of 28-unit-cell-thick $(Pr_{0.85}Y_{0.15})_{0.7}Ca_{0.3}CoO_{3-\delta}$ films on (**g**) YAO(101) and (**h**) LSAT(001). (**i**) "Out-of-plane strain" ($\varepsilon_{zz}$) vs. "in-plane strain" ($\varepsilon_{xx}$) for the $(Pr_{0.85}Y_{0.15})_{0.7}Ca_{0.3}CoO_{3-\delta}$ films in (**a–f**). A bulk pseudocubic lattice parameter of 3.780 Å is assumed, and the dashed line is a straight-line fit corresponding to a Poisson ratio of 0.27. (**j**) Atomic force microscopy image (1.9 × 1.4 μm²) of a 33-unit-cell-thick $(Pr_{0.85}Y_{0.15})_{0.7}Ca_{0.3}CoO_{3-\delta}$ film on YAO, with the extracted RMS roughness ($\sigma_{RMS}$) shown.

LAO, SLGO, LSAT, and STO substrates, using high-pressure-oxygen sputter deposition (see Methods for details). Figure 1 focuses on the specific composition $(Pr_{0.85}Y_{0.15})_{0.7}Ca_{0.3}CoO_{3-\delta}$, i.e., $x = 0.30$, $y = 0.15$, which we generically denote "PYCCO". This specific composition is the focus of the first part of this paper as it well represents an intensively studied bulk compositional regime. The bulk pseudocubic lattice parameter at this composition is 3.780 Å[24], meaning that the lattice mismatches in Fig. 1a-f vary from −2.10% (compressive) on YAO to +3.31% (tensile) on STO, as indicated in each panel. The data support single-phase epitaxial films (confirmed by wider-range XRD, e.g., Fig. S1a), the Laue fringes additionally indicating low surface/interface roughness, particularly under compression (e.g., on YAO in Fig. 1a). The most important feature, however, is the clear shift (dashed line) of the 002 PYCCO film peak to higher angle with increasing tensile strain, evidencing the expected out-of-plane lattice expansion for fully-strained films. This trend abruptly ends on STO, indicating partial (though far from complete) strain relaxation, not unexpected at such extreme strains (+3.31%).

That films on YAO through LSAT are indeed fully-strained to their substrates (i.e., they are pseudomorphic) is illustrated in Fig. 1g, h which show representative asymmetric SXRD reciprocal space maps about the 103 reflections of 28-unit-cell-thick films on YAO and LSAT. In both cases the substrate and primary film reflections occur at identical in-plane lattice vectors, the Laue fringes again being more pronounced under compressive strain (Fig. 1g), indicating somewhat improved epitaxy under compression. Figure 1i plots the resulting "out-of-plane strain" vs. "in-plane strain" components, i.e., $\varepsilon_{zz}$ vs. $\varepsilon_{xx}$. With the exception of the STO case, where partial strain relaxation is clear, the data are well fit by a straight-line through the origin, with slope corresponding to a Poisson ratio of 0.27, typical of cobaltites[48–51]. At this thickness (~30 unit cells), PYCCO films fully strained to their substrates are thus established between −2.10% (on YAO) and +2.34% (on LSAT), although indications of strain relaxation do arise at higher thickness. We note as an aside that the lobes flanking the primary film reflection in Fig. 1g have been observed

before in epitaxial cobaltite films and have been associated with in-plane twin domain structures[51–53].

Turning to microscopy, Fig. 1j shows a representative AFM image from a 33-unit-cell-thick film on YAO, indicating low root-mean-square roughness of 1.4 Å over 1.9 × 1.4 μm². These films are thus very smooth, with closer analysis (Fig. S1c, d) confirming unit-cell-high steps and terraces. Figure 2 then summarizes findings from STEM/EDX, starting with high-angle annular-dark-field (HAADF) STEM images on representative YAO and LAO substrates in Fig. 2a, b, i.e., under −2.10% and +0.25% strain. The images indicate high epitaxial quality and nominally sharp substrate/film interfaces, although some disorder is discernible in the form of local (few-unit-cell-scale) HAADF intensity variations (see Fig. S2 for additional data). This is illustrated quantitatively in Fig. 2e, f, which include line scans of the HAADF intensity along the atomic columns highlighted with the dotted orange lines in Fig. 2a, b. These line scans show variations in the peak intensities, particularly under tension (Fig. 2f). The origin of these intensity variations is clarified by the STEM-EDX images in Fig. 2c, d, which show the spatial distribution of Pr, Co, Ca, and Y in the areas indicated by the teal and green boxes in Fig. 2a, b. While Co appears uniform and the Y EDX intensity is low due to the relatively low Y concentration, the Pr, and particularly Ca maps, reveal noticeable inhomogeneity, again strongest under tension. This is quantified in Fig. 2e, f, where the lower panels compare Pr, Y, and Ca intensities along the atomic columns indicated by the orange dotted lines in Fig. 2a-d. As can be seen particularly clearly in Fig. 2f, atomic columns with low HAADF intensity (vertical grey bands) are consistently associated with low Pr intensity and high Ca intensity. The local disorder evident in Fig. 2a-d thus originates in some level of local Ca-doping fluctuations, which are stronger under tensile strain. These deductions are reinforced through a much larger-scale statistical comparison of HAADF intensities and local compositions in Fig. S3. In addition, we note that Fig. 2 and S2, reveal no evidence of superstructures from, e.g., oxygen vacancy ordering. Such superstructuring is common in strained epitaxial

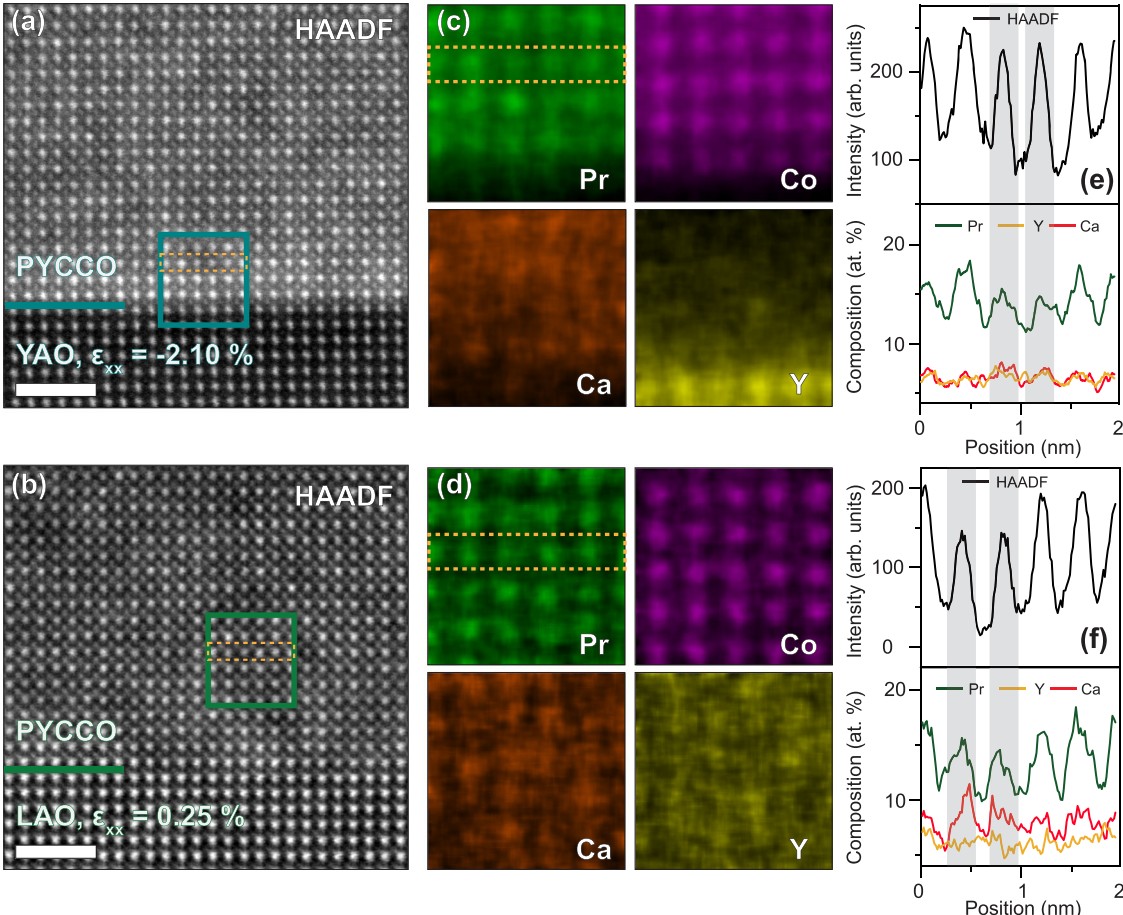

**Fig. 2 | Scanning transmission electron microscopy/energy-dispersive X-ray spectroscopy (STEM/EDX) characterization of $(Pr_{0.85}Y_{0.15})_{0.7}Ca_{0.3}CoO_{3-\delta}$ films.** Room-temperature STEM/EDX characterization of films on YAO ($\varepsilon_{xx} = -2.10$ %) and LAO ($\varepsilon_{xx} = 0.25$ %) substrates (where $\varepsilon_{xx}$ is the "in-plane strain"). Atomic-resolution high-angle annular-dark-field (HAADF) STEM images on (**a**) YAO(101) and (**b**) LAO(001) substrates. The scale bars are 2 nm and the horizontal lines mark the interfaces. (**c, d**) EDX maps of Pr $L\alpha$ (green), Co $K\alpha$ (magenta), Ca $K\alpha$ (orange) and Y $L\alpha$ (yellow) intensities in the boxed regions in (**a,b**) (on YAO and LAO, respectively). (Note that due to the low Y content of these films, the Y values are subject to significant uncertainty). (**e,f**) Line scans of the HAADF intensity (top) and normalized elemental atomic percentages (bottom) along the atomic columns highlighted in (**a-d**) (on YAO and LAO, respectively). The grey vertical bands highlight columns with larger HAADF intensity and compositional fluctuations, as discussed in the text.

cobaltite films[54–59] but was not detected here, for reasons that are not entirely clear.

With high-quality, epitaxial, smooth PYCCO films established, Fig. 3 moves to electronic and magnetic properties vs. heteroepitaxial strain. Shown here are the $T$ dependencies of $\rho$ (top panels) and $M$ (bottom panels), the latter measured in a 100 Oe (10 mT) in-plane magnetic field after field cooling in 10 kOe (1 T), for representative ~30-unit-cell-thick films on YAO, SLAO, LAO, and LSAT. Starting with SLAO/PYCCO, under mild compression ($\varepsilon_{xx} = -0.63$%), $M(T)$ (Fig. 3f) shows no obvious evidence of ferromagnetism, broadly consistent with bulk behavior at this composition ($(Pr_{0.85}Y_{0.15})_{0.7}Ca_{0.3}CoO_{3-\delta}$)[18,22–24,27]. Consistent with the prior reports discussed in the Introduction[46,47], $\rho(T)$ on this substrate is rather indeterminate: $\rho(300$ K) is low (~1 mΩcm) and weakly $T$-dependent, but a broad, weak, metal-insulator transition is evidenced near the bulk $T_{vt}$ of 130 K, resulting in insulating behavior at low $T$. Zabrodskii analysis[60] (Fig. S4c) confirms a metal-insulator transition centered at $T_{vt} \approx 132$ K, although this is broad and diffuse, consistent with prior work[46,47]. The exact origin of this broadening is not entirely clear (it may be related to the aforementioned nanoscale doping inhomogeneity), although we demonstrate below that it is eliminated under stronger compression.

More interesting results emerge under tension. As can be seen from Fig. 3g, h, for example, clear evidence of ferromagnetism emerges under tensile strain, with $T_C$ increasing from ~53 to ~74 K from LAO (+0.25% strain) to LSAT (+2.34% strain). Correspondingly, Fig. 3c, d reveal relatively low resistivity with weak $T$ dependence, and no $T$-dependent metal-insulator transition. Closer inspection (Fig. S5) in fact reveals clear changes in behavior in $\rho(T)$ near $T_C$ in these cases, consistent with a weakly-metallic F state under tension, the increased resistivity on LSAT likely being related to defects and disorder at the highest tensile strains[50]. It should be emphasized here that at this composition ($(Pr_{0.85}Y_{0.15})_{0.7}Ca_{0.3}CoO_{3-\delta}$) no ferromagnetic or metallic state exists in bulk[18,22–24,27]. Y substitutions below ~0.05 are in fact required to enter a F metallic phase, the $T_C$ even at $y = 0$ (i.e., in $Pr_{0.7}Ca_{0.3}CoO_{3-\delta}$) being only ~75 K[16,45]. Qualitatively consistent with the prior report of Padilla-Pantoja et al. on $Pr_{0.5}Ca_{0.5}CoO_{3-\delta}$[44], tensile strain is thus highly efficient at stabilizing F metallic behavior in PYCCO. For clarity, we also explicitly note here that we describe the behavior in Fig. 3c, d (and at high $T$ in Fig. 3a, b), as "metallic" in the sense that $\rho$ is of the order of 1 mΩcm (similar to bulk PYCCO in the metallic region of its phase diagram[22,24,61]) and the $T$ dependence is sufficiently weak that finite conductivity is indicated as $T \rightarrow 0$, even though $d\rho/dT$ is negative. This is consistent with the decrease seen at low $T$ in Figs. S5(b,d), and the marginally metallic state typically seen in bulk PYCCO in the "metallic" regions of its phase diagram[22,24,61].

The behavior under strong compressive strain is yet more striking, and is in fact a central finding of this work. As shown in Fig. 3a, e,

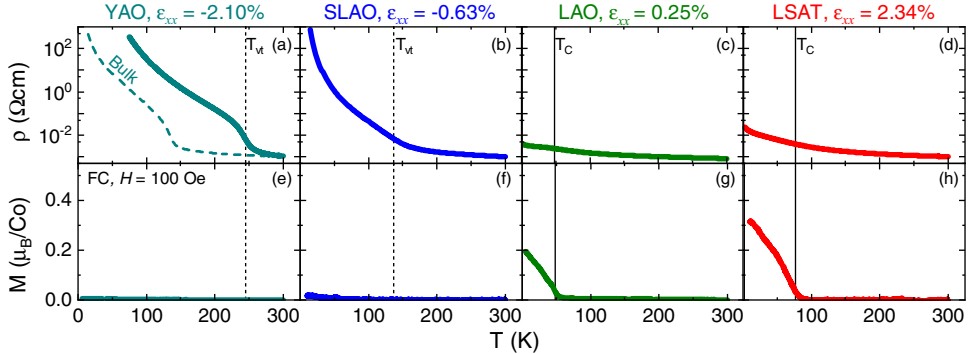

**Fig. 3 | Strain-dependent electronic and magnetic properties of $(Pr_{0.85}Y_{0.15})_{0.7}Ca_{0.3}CoO_{3-\delta}$ films.** Temperature ($T$) dependence of the resistivity ($\rho$) (top panels, $\log_{10}$ scale, taken on warming) and magnetization ($M$) (bottom panels) of 28- to 33-unit-cell-thick $(Pr_{0.85}Y_{0.15})_{0.7}Ca_{0.3}CoO_{3-\delta}$ films on (**a,e**) YAO(101), (**b,f**) SLAO(001), (**c,g**) LAO(001), and (**d,h**) LSAT(001) substrates. (In (**a**), a bulk poly-crystalline sample at the same composition is shown for reference (dashed curve)). The "in-plane strains" $\varepsilon_{xx}$ are shown. $M$ was measured in-plane in a 100 Oe (10 mT) applied field ($H$) after field-cooling in 10 kOe (1 T). Vertical dashed and solid lines mark the valence transition temperature ($T_{vt}$) and Curie temperature ($T_C$), respectively; $T_{vt}$ is determined from the peaks in Zabrodskii plots[60], as shown in Fig. S4(c).

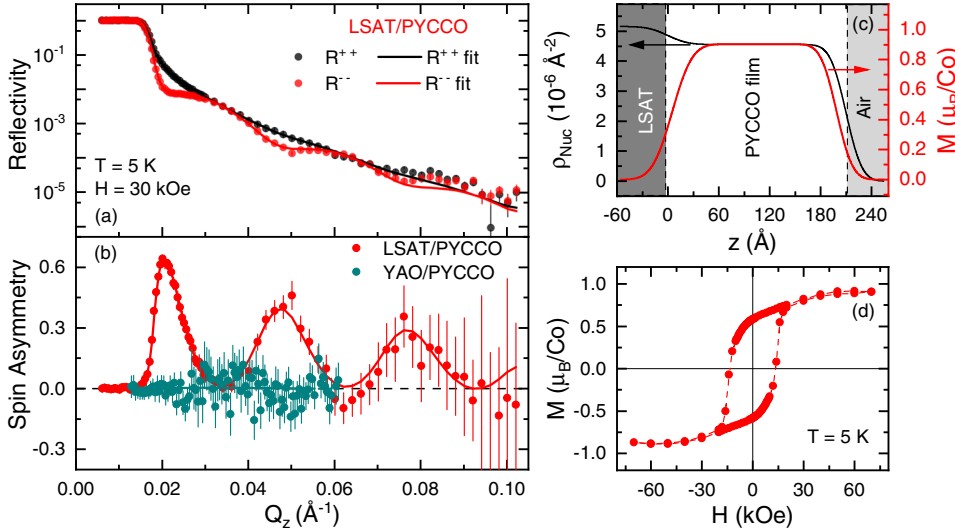

**Fig. 4 | Polarized neutron reflectometry (PNR) characterization of ferromagnetism in $(Pr_{0.85}Y_{0.15})_{0.7}Ca_{0.3}CoO_{3-\delta}$ films.** (**a**) Neutron reflectivity ($R$) vs. scattering wave vector magnitude ($Q_z$) from a 56-unit-cell-thick $(Pr_{0.85}Y_{0.15})_{0.7}Ca_{0.3}CoO_{3-\delta}$ film on LSAT(001) at 5 K, in a 30-kOe (3-T) in-plane magnetic field ($H$). Black and red points denote the non-spin-flip channels $R^{++}$ and $R^{--}$, respectively, and the solid lines are the fits discussed in the text. (**b**) Spin asymmetry [SA = $(R^{++}\text{-}R^{--})/(R^{++}+R^{--})$] vs. $Q_z$ extracted from (**a**), along with the same for a 28-unit-cell-thick film on YAO(101). (**c**) Depth ($z$) profiles of the nuclear scattering length density ($\rho_{Nuc}$, left-axis) and magnetization ($M$, right-axis) extracted from the fits to the LSAT(001) data shown in (**a, b**). (**d**) 5-K $M$ vs. $H$ hysteresis loop from a 56-unit-cell-thick $(Pr_{0.85}Y_{0.15})_{0.7}Ca_{0.3}CoO_{3-\delta}$ film on LSAT(001). Error bars on the neutron data are ±1 standard deviation.

2.10% compressive heteroepitaxial strain on YAO(101) not only eliminates any signature of F behavior but also generates a notably sharp metal-insulator transition at a strongly enhanced $T_{vt} \approx 245$ K (that this metal-insulator transition is indeed a valence transition of the type seen in bulk is directly verified below). Recall here that $T_{vt}$ in the $Pr_{1-x}Ca_xCoO_{3-\delta}$ system peaks at ~90 K at $x = 0.50$[15–20], while $T_{vt}$ in the $(Pr_{1-y}Y_y)_{1-x}Ca_xCoO_{3-\delta}$ system is ~135 K at $x = 0.30$, $y = 0.15$[18,22–24,27], meaning that the result in Fig. 3a constitutes a ~90% $T_{vt}$ enhancement over the same composition in bulk (dashed line in Fig. 3a), and an ~170% enhancement over bulk $Pr_{1-x}Ca_xCoO_{3-\delta}$. As quantified by Zabrodskii analysis[60] (Fig. S4c), the $T$-dependent metal-insulator transition in Fig. 3a is also as sharp as in bulk $(Pr_{1-y}Y_y)_{1-x}Ca_xCoO_{3-\delta}$, and is accompanied by weak thermal hysteresis (Fig. S4a, b, d), confirming the first-order nature of the transition in compressively-strained films. It must be emphasized that while the thermal hysteresis shown in Fig. S4 is modest, this is also true of bulk, where the hysteresis width is reported to be well under 1 K at such compositions[62]. As returned to below, where we demonstrate that YAO/PYCCO films also undergo

simultaneous structural and valence transitions, the metal-insulator transition under large compressive strain on YAO is thus bulk-like, but with strongly enhanced $T_{vt}$.

Prior to probing the detailed nature of the transition in Fig. 3a, we first verify that the ferromagnetism under tensile strain is indeed long-range-ordered and depth-wise-uniform. This was achieved via PNR measurements on a representative LSAT/PYCCO film, as shown in Fig. 4a (see Methods for further details). Under these conditions (5 K in an applied magnetic field of 30 kOe (equivalent to 3 T)), the non-spin-flip reflectivities $R^{++}$ and $R^{--}$ (where the "+" and "−" indicate the neutron spin orientation of the incoming and outgoing beams relative to the sample magnetization) exhibit clear splitting vs. the magnitude of the specular scattering wave vector $Q_z$, directly evidencing long-range F order. As shown in Fig. 4b, the resulting spin asymmetry, SA = $(R^{++}\text{-}R^{--})/(R^{++}+R^{--})$, exhibits prominent oscillations in LSAT/PYCCO films. A standard refinement process (see Methods and Table S1 for details) results in the solid line fits through the $R^{++}$ and $R^{--}$ data in Fig. 4a and the spin asymmetry in Fig. 4b, resulting in the

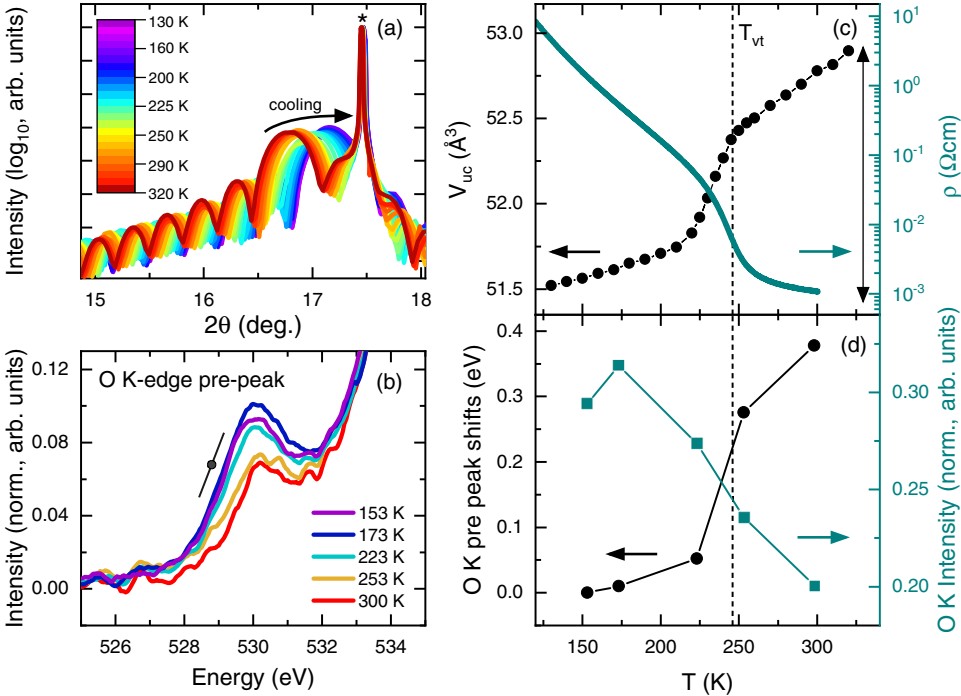

**Fig. 5 | Temperature-dependent structural and valence transitions in compressively-strained (Pr$_{0.85}$Y$_{0.15}$)$_{0.7}$Ca$_{0.3}$CoO$_{3-\delta}$ films.** (**a**) Temperature (*T*)-dependent synchrotron X-ray diffraction (SXRD) around the 002 film peak of a 26-unit-cell-thick (Pr$_{0.85}$Y$_{0.15}$)$_{0.7}$Ca$_{0.3}$CoO$_{3-\delta}$ film on YAO(101) from 130 K to 320 K (see color scale). (**b**) Electron energy loss spectra (EELS) near the O *K*-edge pre-peak region of a ~30-unit-cell-thick (Pr$_{0.85}$Y$_{0.15}$)$_{0.7}$Ca$_{0.3}$CoO$_{3-\delta}$ film on YAO(101) from 153 K to 300 K. The data are normalized to the post-edge intensity (580-585 eV) to enable direct comparison. The black line illustrates the low-*T* slope while the black point marks the low-*T* inflection point. This *T*-dependent inflection point was used to determine the pre-peak shift with *T*, as described in more detail in Fig. S7 and the

associated caption. Right panels: *T*-dependence of the pseudocubic unit cell volume ($V_{uc}$, left axis to (**c**)), resistivity ($\rho$, right axis to (**c**)), O *K*-edge pre-peak shift (left-axis to (**d**), relative to the lowest-*T* (153 K) data point), and O *K* pre-edge intensity (right-axis to (**d**), integrated in the range from 527-532 eV). All data are from ~30-unit-cell-thick (Pr$_{0.85}$Y$_{0.15}$)$_{0.7}$Ca$_{0.3}$CoO$_{3-\delta}$ films on YAO(101). The vertical dashed line in the right panels marks the valence transition temperature $T_{vt}$, as determined from the peak in a Zabrodskii plot[60], as shown in Fig. S4(c). The double-ended vertical black arrow in (**c**) marks the volume change from DFT calculations of Pr$_{0.5}$Ca$_{0.5}$CoO$_3$.

refined depth (*z*) profiles of nuclear scattering length density ($\rho_{Nuc}$) and magnetization in Fig. 4c. $\rho_{Nuc}(z)$ (black line) is relatively unremarkable, suggesting interface and surface roughness/intermixing over ~4 unit cells, with film density within ~1% of the theoretical value (see Table S1). More significantly, the extracted *M*(*z*) in Fig. 4c (red line) directly confirms depth-wise-uniform F magnetization, saturating at 0.90 ± 0.01 μ$_B$/Co. Typical dead layer formation is found at the interface and film surface, amounting to ~2 unit cells, comparable to other F cobaltite films[51,63,64]. As shown in Fig. 4d, magnetization hysteresis loops also support in-plane ferromagnetism, the saturation magnetization of 0.91 ± 0.05 μ$_B$/Co being in excellent accord with the 0.90 ± 0.01 μ$_B$/Co from PNR (which is absolute, and not subject to the substrate background subtraction issues in magnetometry). Magnetometry and PNR data are thus in close agreement on the F metallic state stabilized in PYCCO films by strong tensile strain. Importantly, equivalent PNR measurements on thin PYCCO films under compressive strain, on YAO, rule out ferromagnetism to <0.01μ$_B$/Co, as illustrated by the spin asymmetry data in Fig. 4b (teal points), and the fuller analysis in Fig. S6. This is consistent with the metal-insulator transition in Fig. 3a, i.e., the insulating low-*T* state under compression is non-F, as expected.

Returning to the enhanced metal-insulator transition discovered under large compressive strain (Fig. 3a), Fig. 5 shows the results of *T*-dependent SXRD and STEM/EELS studies probing for accompanying structural, valence, and spin-state transitions. Figure 5a first shows a dense set of *T*-dependent specular SXRD scans around the film 002 reflection of a 26-unit-cell-thick YAO/PYCCO film. A substantial upshift in the film peak position occurs on cooling, other features such as fringes remaining unaffected. The *c*-axis lattice parameter thus shrinks

on cooling, as illustrated in Fig. 5c, where the resulting unit cell volume ($V_{uc}$, left axis) is plotted vs. *T* along with $\rho(T)$ (right axis) for comparison. The correspondence between $\rho(T)$ and $V_{uc}(T)$ is striking, the linear thermal expansion above ~250 K and below ~225 K being interrupted by an abrupt collapse of the cell volume by ~1% over an ~20 K window near $T_{vt}$. The structural transition accompanying the *T*-dependent metal-insulator transition in bulk PYCCO is thus preserved in strongly-compressively-strained thin films (on YAO).

Figure 5b shows corresponding *T*-dependent EELS data in the pre-peak region of the O *K* edge of a YAO/PYCCO film. (Wider range plots of the entire O *K* edge region are shown in Figs. S7a-c, along with reference spectra for YAO and LAO substrates). The pre-peak near 530 eV is well established to be sensitive to the O 2*p* hole density, Co valence, and Co ion spin-state[55,58,65–69], meaning that this feature is an excellent probe of changes in both the Co ion valence and spin-state. The pre-peak position, intensity, and shape are indeed found to be strongly *T*-dependent in the 300-150 K window (Fig. 5b), control experiments on LAO/PYCCO films (0.25% tensile strain) displaying no such effects (compare Figs. S7d and g). While quantitative analysis is complicated by the combined influence of the Co ion spin-state and valence, as well as conflicting reports on the exact impact of Co valence[55,58,66,68,69], the *T*-dependent changes in pre-peak intensity and position in Fig. 5b nevertheless clearly evidence a *T*-dependent valence/spin-state transition. This is reinforced in Fig. 5d, which plots the *T*-dependent shift (relative to the lowest-*T* (153 K) point) of the O *K* edge pre-peak position (left axis), as well as the O *K* edge intensity (right axis). Details on exactly how these parameters were extracted from EELS data are provided in Figs. S7d-f and the associated caption. Strikingly, the pre-peak position shift in Fig. 5d (which amounts to

~0.4 eV) closely mirrors the behavior in $V_{uc}(T)$ in Fig. 5(c), while the intensity rapidly increases on cooling through the transition. These behaviors unambiguously indicate that the $T$-dependent metal-insulator transition seen in Fig. 5c is accompanied not only by a structural transition (Fig. 5a, c) but also a spin-state/valence transition (Fig. 5b, d). As a whole, the data of Fig. 5 thus demonstrate that the enhanced-$T_{vt}$ transition in PYCCO films under strong compressive strain is indeed of the same essential nature as seen in bulk, i.e., it is a coupled valence/structural/spin-state/metal-insulator transition. Future work at the Co and Pr spectroscopic edges could provide additional complementary insight, including fine details of the evolution with strain. As a final important point on these data, we emphasize that in LAO/PYCCO films, under tensile strain, the behavior in Fig. 5a-d is completely absent, with only minor $T$ dependence arising in the O $K$ edge pre-peak region (see Figs. S7f and i).

The results from Figs. 1–5 are summarized in a heteroepitaxial strain "phase diagram" in Fig. 6a, i.e., a $T$-$\varepsilon_{xx}$ plot with $T_{vt}$ and $T_C$

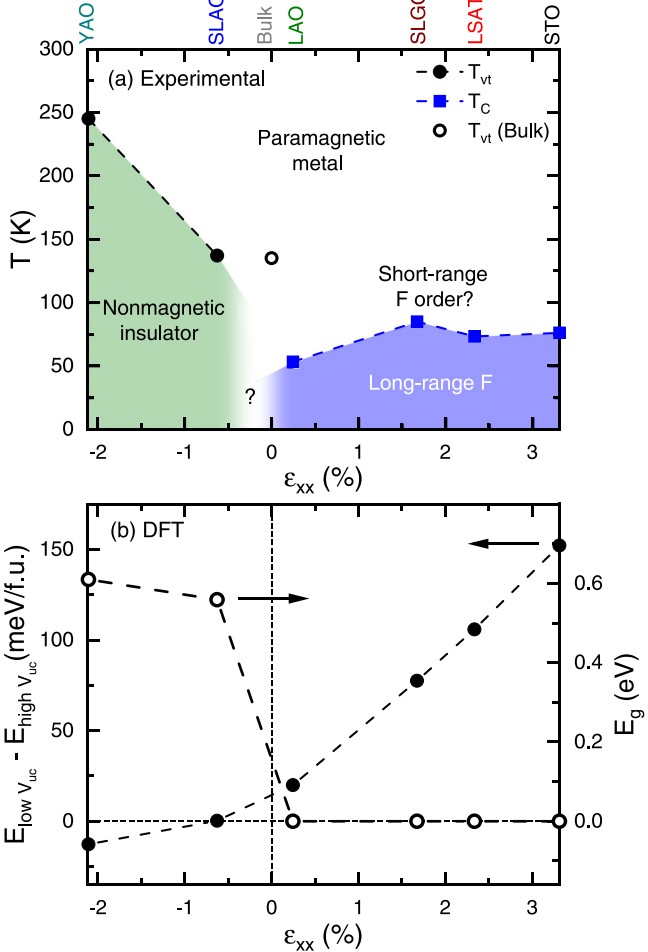

**Fig. 6 | Strain "phase diagram" of $(Pr_{0.85}Y_{0.15})_{0.7}Ca_{0.3}CoO_{3-\delta}$ and DFT results. (a)** Experimental temperature ($T$) vs. "in-plane strain" ($\varepsilon_{xx}$) phase diagram for $(Pr_{0.85}Y_{0.15})_{0.7}Ca_{0.3}CoO_{3-\delta}$. Thin film (solid points, ~30-unit-cell-thick) and bulk (open point) data are shown, with the relevant substrate indicated at the top. (Note that partial strain relaxation occurs on STO (see Fig. 1(f, i), meaning that the true $\varepsilon_{xx}$ value on that substrate only will be lower than shown). The valence transition temperature $T_{vt}$ (circles) and Curie temperature $T_C$ (squares) are plotted. Green, white, and blue phase fields indicate "nonmagnetic insulator", "paramagnetic metal", and "long-range ferromagnet (F)", respectively. (**b**) Energy difference between the low- and high-unit-cell-volume states of $Pr_{0.5}Ca_{0.5}CoO_3$ vs. $\varepsilon_{xx}$ (left axis) as obtained from density functional theory calculations. Note that the absolute energies are both negative. On the right axis is the corresponding energy gap ($E_g$) of the ground-state structure.

marked. The white, green, and blue phase fields here correspond to high-$T$ paramagnetic metal, nonmagnetic insulator, and long-range F regions, respectively, the black circles and blue squares being $T_{vt}$ and $T_C$. At $\varepsilon_{xx} > 0$ (tensile strain) the dominant feature is the transition on cooling from paramagnetic metal to F metal, likely preceded, based on bulk findings[24,45], by weak, shorter-range F order. This behavior persists to the smallest tensile strain probed here (+0.25%). At $\varepsilon_{xx} < 0$ (compressive strain), on the other hand, the dominant feature is the transition on cooling from paramagnetic metal to nonmagnetic insulator, through the first-order coupled metal-insulator/spin-state/structural/valence transition at $T_{vt}$. As already emphasized, $T_{vt}$ is rapidly enhanced with increasing magnitude of compressive strain, reaching 245 K at −2.10%, i.e., approximately double that of the same composition in bulk, which is marked by the open circle in Fig. 6. Such strong sensitivity of the electronic and magnetic ground state to strain in PYCCO raises the obvious question of whether the chemical pressure exerted by Y is actually needed to realize strain enhancement of $T_{vt}$. We thus repeated the measurements in Fig. 3 for $Pr_{0.7}Ca_{0.3}CoO_{3-\delta}$ films (i.e., $y = 0$ PYCCO) grown on YAO, SLAO, LAO, and LSAT, resulting in the $\rho(T)$ and $M(T)$ shown in Fig. S8, and the strain phase diagram in Fig. S9. The result is a phase diagram shifted to more negative $\varepsilon_{xx}$, with maximum $T_{vt} \approx 140$ K on YAO. (Recall that in the bulk at this composition, $T_{vt} = 0$, as no valence transition occurs until $x = 0.5$.) The strong enhancement of $T_{vt}$ with compressive strain thus persists, simply with a lower maximum $T_{vt}$ (140 K for $Pr_{0.7}Ca_{0.3}CoO_{3-\delta}$ films compared to 245 K for $(Pr_{0.85}Y_{0.15})_{0.7}Ca_{0.3}CoO_{3-\delta}$ films), as might be expected. Complete strain tuning between an F metallic ground state under tension and a stabilized valence/spin-state/structural/metal-insulator transition under compression is thus a robust result, independent of exact composition. Interestingly, while $(Pr_{0.85}Y_{0.15})_{0.7}Ca_{0.3}CoO_{3-\delta}$ displays higher $T_{vt}$ under compression (we also studied various other $x$ and $y$ in $(Pr_{1-y}Y_y)_{1-x}Ca_xCoO_{3-\delta}$ films under compression on YAO (see Fig. S10)), Y-free $Pr_{0.7}Ca_{0.3}CoO_{3-\delta}$ exhibits the sharpest metal-insulator transition, with strongest thermal hysteresis (Fig. S10b, c), potentially due to better chemical homogeneity.

Seeking further insight into the tensile and compressive strain stabilization of F metallic and nonmagnetic insulating phases, respectively, we also performed strain-dependent DFT calculations of the relatively simple $Pr_{0.5}Ca_{0.5}CoO_3$ system. As noted in the Introduction, DFT is capable of reproducing the valence/spin-state transition in bulk $Pr_{0.5}Ca_{0.5}CoO_3$; this was achieved by imposing the experimental high-$T$ $V_{uc}$ to model the high-$T$ phase, and the experimental collapsed $V_{uc}$ to model the low-$T$ phase[17]. We extended this method to hetero-epitaxial strain tuning by taking the bulk structures in the metallic (high $V_{uc}$) and insulating (low $V_{uc}$) states[17], constraining the in-plane lattice parameters to those of the experimental substrates, and then relaxing the atomic positions and out-of-plane lattice parameters in DFT + $U$ (see Methods for further details). Using the occupation control method of Ref. 70, we ensured that the system remains in the defined $V_{uc}$/valence/spin-state throughout the relaxation, resulting in the volume difference (after relaxation) indicated by the black double-ended arrow in Fig. 5c. The central results from such calculations are shown in Fig. 6b, which plots the $\varepsilon_{xx}$ dependence of the difference in total energy per formula unit of $Pr_{0.5}Ca_{0.5}CoO_3$ between the low- and high-$V_{uc}$ states ($E_{low\ Vuc} - E_{high\ Vuc}$, left axis) and the energy gap of the ground-state structure ($E_g$, right axis). The results strongly support the experimental observations: The high-$V_{uc}$ state is significantly lower in energy than the low-$V_{uc}$ state under tensile biaxial strain (i.e., ($E_{low\ Vuc} - E_{high\ Vuc}$) is positive for positive $\varepsilon_{xx}$, but ($E_{low\ Vuc} - E_{high\ Vuc}$) gradually decreases towards compressive strains (negative $\varepsilon_{xx}$), eventually inverting. At the same time, the $E_g$ of the ground-state structure transitions from zero at positive $\varepsilon_{xx}$, to >100 meV at negative $\varepsilon_{xx}$. This corresponds to the stabilization of the metallic F state under tension and the insulating non-F state under compression, exactly as in Fig. 6a for $(Pr_{0.85}Y_{0.15})_{0.7}Ca_{0.3}CoO_{3-\delta}$ and Fig. S9 for $Pr_{0.7}Ca_{0.3}CoO_{3-\delta}$. While

various factors will impact the exact energies in Fig. 6b, including the $U$ values imposed on Co and Pr (see Methods and Fig. S11), and the exact Ca and Y composition, the overall trend is robust across various parameter choices (Fig. S11), strongly supporting the experimental observations in Fig. 6a and Fig. S9. In addition to the energies, we also find qualitative agreement between the structural changes observed across the thermal valence transition in $Pr_{0.5}Ca_{0.5}CoO_3$ and the structural changes predicted by DFT vs. strain (see Fig. S12). This is further evidence of the similarity between thermal and strain-tuned valence transitions in these systems. Fig. S13 shows the full strain evolution of the density-of-states in the high- and low-$V_{uc}$ states, further supporting Fig. 6(b), particularly the $E_g(\varepsilon_{xx})$ behavior.

Finally, motivated to promote $T_{vt}$ yet further, to ambient temperature, we also studied higher-Y-content $(Pr_{1-y}Y_y)_{1-x}Ca_xCoO_{3-\delta}$, i.e., different compositions to the above. Bulk polycrystalline samples were thus prepared with $x = 0.25$, 0.30, and 0.40, with variable $y$. As shown in Fig. 7a, due to the similarity of the $Pr^{3+}$ and $Ca^{2+}$ ionic radii, the $T_{vt}$ of such samples is essentially controlled by the Y substitution fraction only (i.e., $y(1-x)$). Similar to other bulk reports[22,24,27,28,61], a practical limit on $T_{vt}$ is reached, in this case at ~180 K, due to precipitation of a $Y_2O_3$ secondary phase (as detected in XRD). Epitaxial films with $x = 0.30$, $y = 0.25$ were thus prepared (this composition is highlighted in Fig. 7a and contrasted with the $x = 0.30$, $y = 0.15$ emphasized thus far in this paper), focusing on YAO substrates, i.e., large compressive strain. Fig. S14 first confirms that such films are of comparable quality to others in this work, based on specular XRD (Fig. S14a), rocking curves (Fig. S14b), and AFM (Fig. S14d) measurements. Figure 7b then compares the $\rho(T)$ of such films to $y = 0.15$ films on YAO (as in Fig. 3a), as well as bulk polycrystals at the same compositions. The strong enhancement of $T_{vt}$ under compressive strain (see the horizontal arrows in Fig. 7b) is maintained at $y = 0.25$, $T_{vt}$ increasing from 179 K in bulk to 291 K under compression on YAO (see Fig. S10c for the corresponding Zabrodskii plot), i.e., by ~100 K. Significantly, Fig. 7c further demonstrates that the transition centered at 291 K in compressively-strained $x = 0.30$, $y = 0.25$ films is indeed clear in $T$-dependent XRD also (see Fig. S14c for raw data), the correspondence between $V_{uc}(T)$ (left axis of Fig. 7c) and $\rho(T)$ (right axis of Fig. 7c) being striking, just as in Fig. 5c. Strain stabilization of the $(Pr_{1-y}Y_y)_{1-x}Ca_xCoO_{3-\delta}$ coupled valence/structural/spin-state/metal-insulator transition to room temperature is thus confirmed. The transition seen in Fig. 7b, c in the $x = 0.30$, $y = 0.25$ case is in fact not entirely complete until at least ~330 K.

## Discussion

This work reports the first study of the influence of wide-ranging compressive and tensile biaxial strain on high-quality epitaxial thin films of $(Pr_{1-y}Y_y)_{1-x}Ca_xCoO_{3-\delta}$ perovskite cobaltites. Complete control over the electronic and magnetic ground state has been achieved, from ferromagnetic and metallic under tension, to nonmagnetic and insulating under compression, reproduced by DFT calculations. In particular, the unique valence-driven first-order coupled structural/spin-state/metal-insulator transition in this material system has been strain-stabilized under large compression to at least 291 K. This brings a valence transition in a perovskite oxide from the cryogenic temperatures in bulk, to room temperature, realizing not only a metal-insulator transition rivaling classic systems such as vanadium oxides[1,21], but also additional function stemming from valence and spin-state control, with abundant device potential. The latter could include resistive switching and thresholding devices, switchable magnetic, optical, and photonic devices, neuromorphic computing elements, etc.

Strain phase diagrams such as those in Fig. 6a and S9 also generate fundamental opportunities. In particular, there are two clear ways that the long-range ferromagnetic metal and nonmagnetic insulator phase fields could evolve and connect in the intermediate $\varepsilon_{xx}$ region between −0.63 and 0.25% in Fig. 6a. One could envision a first-order transition vs. strain, for example, with phase coexistence of nonmagnetic

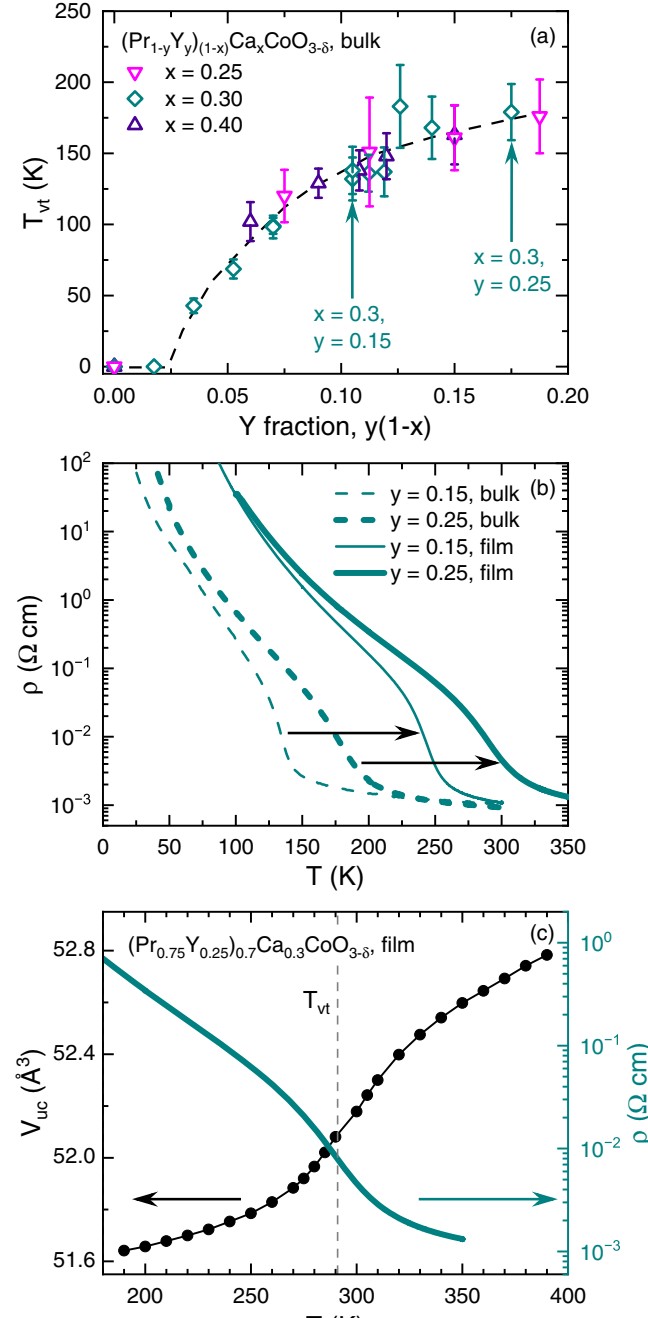

**Fig. 7 | Room-Temperature $T_{vt}$ in compressively-strained $(Pr_{0.75}Y_{0.25})_{0.7}Ca_{0.3}CoO_{3-\delta}$.** (a) $T_{vt}$ vs. Y substitution fraction ($y(1-x)$) for bulk polycrystalline $(Pr_{1-y}Y_y)_{1-x}Ca_xCoO_{3-\delta}$ with $x = 0.25$, 0.30, and 0.40. The $x = 0.30$, $y = 0.15$ and $x = 0.30$, $y = 0.25$ compositions are highlighted, i.e., the standard composition in this work, and the highest-$T_{vt}$ composition, respectively. Error bars correspond to the approximate width of the transition from the widths of the peaks in Zabrodskii plots (see Figs. S4, S10), and the dashed line is a guide to the eye. (b) Temperature ($T$) dependence of the resistivity ($\rho$) ($\log_{10}$ scale, taken on warming) of $x = 0.30$ samples at $y = 0.15$ and 0.25, for both bulk polycrystals (dashed) and 30-40-unit-cell-thick films on YAO(101) substrates (solid). Horizontal arrows highlight the strain enhancement on the transition temperature, which reaches 291 K at $y = 0.25$ (see Fig. S10). (c) $T$-dependence of the pseudocubic unit cell volume ($V_{uc}$, left axis) and resistivity ($\rho$, right axis) of a 41-unit-cell-thick $(Pr_{0.75}Y_{0.25})_{0.7}Ca_{0.3}CoO_{3-\delta}$ film on YAO(101). The vertical dashed line marks the valence transition temperature $T_{vt} = 291$ K, as determined from the peak in a Zabrodskii plot[60], as shown in Fig. S10.

insulating and ferromagnetic metallic phases, with $T$-dependent competition. Alternatively, one could also envision a quantum critical point, where the long-range ferromagnetic phase is extinguished by $T_C$ being driven to $T = 0$, at which point $T_{vt}$ could rise from $T = 0$ to the ~132 K at $\varepsilon_{xx} = -0.63\%$. A related scenario has in fact been recently hypothesized in bulk LaCoO$_3$ tuned via Sc substitution[71]. Strained PYCCO films could serve as a model system to probe such emergent physics, one attractive approach (in addition to controlled strain relaxation with increased thickness) being to apply continuously-tunable uniaxial compression to samples such as tensile-strained LAO/PYCCO, just on the ferromagnetic side of the transition (Fig. 6a). Uniaxial pressure cells have undergone substantial advancements recently, and can now generate several percent compressive strain[72,73], raising the possibility of combining discrete heteroepitaxial strain with continuously tunable compression of underlying substrates. As illustrated by comparing Fig. 6a and S9, the (Pr$_{1-y}$Y$_y$)$_{1-x}$Ca$_x$CoO$_{3-\delta}$ system can be composition-tuned to generate low $T_C$ under small tensile strain, providing an ideal starting point from which to continuously tune with external compression, tracking $T_C$, $T_{vt}$, and the nature of the transitions. The applied implications of stabilizing room-temperature valence transitions in complex oxides are thus accompanied by exciting fundamental prospects, and we thus anticipate substantial further research on strain-tuned Pr-based epitaxial cobaltites.

## Methods

### Thin film deposition and structural/chemical characterization

Nominally-stoichiometric polycrystalline PYCCO sputtering targets (2" (5 cm) diameter) were synthesized by solid-state reaction, cold pressing, and sintering, from Pr$_6$O$_{11}$, Y$_2$O$_3$, CaCO$_3$, and Co$_3$O$_4$ powders[24]. These targets were then used for high-pressure-oxygen sputter deposition of PYCCO films (26 unit cell < $t$ < 56 unit cell), under similar conditions to La$_{1-x}$Sr$_x$CoO$_{3-\delta}$[49–51,63,74,75]. Briefly, YAO, SLAO, LAO, SLGO, LSAT, and STO substrates were annealed at 900 °C in 1 Torr (133 Pa) of flowing ultrahigh-purity O$_2$ (99.998%) for 15 mins prior to growth. Deposition then took place at 600 °C substrate temperature, 25-35 W of DC sputter power, and 1.35 Torr (180 Pa) of flowing O$_2$, resulting in 3.5–6.0 Å min$^{-1}$ growth rates; postgrowth, films were cooled to ambient in 600 Torr (80 kPa) of O$_2$ at -15 °C/min. Bulk PYCCO($x = 0.3$, $y = 0.15$) crystallizes in the *Pnma* space group with 300 K pseudocubic lattice parameter ($a_{pc} = \frac{1}{2}\sqrt{a_{orth}^2 + c_{orth}^2}$) of 3.780 Å, leading to compressive strain on YAO (−2.11%) and SLAO (−0.63%), and tensile strain on LAO (0.25%), SLGO (1.67%), LSAT (2.34%), and STO (3.31%). With respect to O stoichiometry, we note that: (*i*) the growth and cooling conditions were kept constant for all films; (*ii*) the phase diagram in Fig. 6a shows a metallic phase under tension (contrary to the generally expected trend of higher O vacancy concentration under tensile strain); and (*iii*) O vacancy formation energies in cobaltite films in fact respond approximately symmetrically[75] to positive/negative strains. These points (and many others) rule out that the phase behavior in Fig. 6a is driven by O stoichiometry effects, instead being fundamentally strain-driven.

Film thicknesses, depth profiles, and surface roughnesses were determined by grazing incidence X-ray reflectivity using a lab X-ray (Cu $K_\alpha$) source in a Rigaku Smartlab XE diffractometer. Surface morphology was additionally studied with contact-mode AFM in a Bruker Nanoscope V Multimode 8. As discussed below, SXRD characterization was performed at the Advanced Photon Source (APS) in both specular and RSM modes. This was done on the 12-ID-D beamline, using 22 keV ($\lambda = 0.564$ Å) radiation, a six-circle Huber goniometer, and a Pilatus II 100 K area detector. Cross-section samples for STEM were prepared in an FEI Helios Nanolab G4 dual-beam Focused Ion Beam microscope with 30 keV Ga ions. Amorphous C was first deposited on the PYCCO films to protect from damage by the ion beam. Further ion milling employed a 2 keV Ga ion beam to remove surface damaged layers from thinned specimens. HAADF-STEM imaging, EDX spectroscopic imaging, and room temperature EELS were then performed in an

aberration-corrected FEI Titan G2 60-300 (S)TEM, equipped with a CEOS DCOR probe corrector, monochromator, super-X EDX spectrometer, and Gatan Enfinium ER EELS spectrometer. The microscope was operated at 200 keV, with a probe current of 100 pA for imaging and 150 pA for EDX acquisitions. HAADF-STEM images were acquired with a probe convergence angle of 25.5 mrad and detector inner and outer collection angles of 55 mrad and 200 mrad, respectively. Atomic-resolution EDX maps were acquired and analyzed using Bruker Esprit software; drift correction was performed live on the software. For EELS, the collection angle was 14 mrad and an energy dispersion 0.05 eV/channel was used in Dual EELS mode. The full-width at half-maximum (FWHM) of the zero-loss peak was 1 eV during acquisitions.

### Electronic and magnetic measurements

Magnetometry was done in a Quantum Design Magnetic Property Measurement System from 5-300 K in in-plane magnetic fields to 70 kOe (7 T). Electronic transport measurements were performed in a Quantum Design Physical Property Measurement System from 2 to 300 K, using a Keithley 2400 source-measure unit. Indium contacts were employed, in a four-terminal van der Pauw geometry, carefully selecting excitation currents to avoid nonohmicity and self-heating.

### Polarized neutron reflectometry

PNR data were taken on both the Polarized Beam Reflectometer (PBR) at the NIST Center for Neutron Research and the Beamline 4A Magnetism Reflectometer (MR) at the Spallation Neutron Source, Oak Ridge National Laboratory. Saturating in-plane magnetic fields of 30 kOe (3 T) were applied in all cases, and data were typically taken at 5 K, 200 K, and 300 K. Our fitting procedure included data at all available temperatures, keeping all but the magnetic parameters the same at each temperature. On PBR, a 4.75 Å monochromated and polarized incident beam was used, along with full polarization analysis. On MR, 30 Hz polarized neutron beam pulses were used with wavelength centered around 7.5 Å (see also Fig. S6). $R^{++}$ and $R^{--}$ were measured vs. $Q_z$, where the "+" and "−" denote whether the incident (and reflected) neutron spin polarization is parallel or antiparallel to the magnetic field at the sample. The data were then fit to a slab model using the Refl1D software package[76]; model parameter uncertainties (2$\sigma$) were calculated using the DREAM Markov chain Monte Carlo algorithm.

### Temperature-dependent synchrotron X-ray diffraction

As already noted, SXRD was carried out on the 12-ID-D beamline of the APS, using 22 keV ($\lambda = 0.564$ Å) radiation, a six-circle Huber goniometer, and a Pilatus II 100 K area detector. Temperature-dependent SXRD measurements, from 130-320 K, employed an Oxford Instruments liquid-N$_2$-flow cryocooler.

### Temperature-dependent electron energy loss spectroscopy

Temperature-dependent STEM-EELS measurements were performed in a Thermo Fisher TALOS F200C microscope equipped with a Gatan Enfinium SE EELS spectrometer. The microscope was operated at 200 keV with a probe current of 0.56 nA; the probe convergence angle was 9 mrad, with HAADF detector inner and outer collection angles of 93 and 200 mrad, respectively. Specimens were first cooled to liquid N$_2$ temperatures using a Gatan 632 cryo transfer holder, and then temperature-dependent EELS spectra were collected on warming, using a Gatan cold-stage controller. All EELS spectra were collected using an energy dispersion of 0.05 eV/channel and a 20.3 mrad EELS collection angle; the FWHM of the zero-loss peak was 0.7 eV during data acquisition. For comparison of the temperature-dependent O $K$-edge spectra, each spectrum was normalized in the 580-585 eV energy window. The position of the O pre-peak at each temperature was then determined from the inflection point of the pre-peak slope. Specifically, the EELS spectra were first numerically differentiated, using

30-pixel averaging (a 1.5 eV energy range) to reduce noise. To obtain the inflection point, a second order polynomial was then fitted to the differentiated data, e.g., in the energy window 530.5 to 532.0 eV. This energy window was adjusted to higher or lower energies to accommodate the shift of the O pre-peak with temperature. The zeros of the first derivative of the fitted function were then used to estimate the slope inflection point. Further details are provided in Fig. S7 and the associated caption.

## DFT calculations

As noted in the main text, biaxial-strain-dependent calculations were performed by initializing $Pr_{0.5}Ca_{0.5}CoO_3$ in the high- and low-$V_{uc}$ states, constraining the in-plane lattice parameters to those of the relevant substrate, and then relaxing the atomic positions and out-of-plane lattice parameters in DFT+$U$. For this we used the high- and low-$V_{uc}$ parameters from the 10- and 295-K $Pnma$ structures determined by Fujita et al.[26]; choosing initial structures far from $T_{vt}$, i.e., deep in the high- and low-$V_{uc}$ states was found to help stabilize the states. Following Knizek et al.[17], rock-salt-like arrangements of Pr and Ca were employed in these model calculations of $Pr_{0.5}Ca_{0.5}CoO_3$. To stabilize the desired valences ($Pr^{4+}/Co^{3+}$) in the insulating low-$V_{uc}$ state we constrained the Co density matrix to that of low-spin $Co^{3+}$ during relaxation, using the occupation control method implemented in Ref. 70. The $Co^{3+}$ density matrix for this purpose was extracted from a calculation for undoped $PrCoO_3$, which is $Pnma$ with low-spin $Co^{3+}$ in its ground state. After the calculation for the initial constrained relaxation converged, the relaxation was repeated without imposing the density matrix constraint. Pr occupations were not constrained during this whole process. Similarly, to stabilize the desired valences ($Pr^{3+}/Co^{3.5+}$(intermediate spin)) in the metallic high-$V_{uc}$ state we fixed the $Pr^{3+}$ density matrix to that of $PrCoO_3$ during an initial relaxation. The Co occupation matrix was not constrained during this relaxation as it was found sufficient to simply initialize the Co magnetic moment to 2 $\mu_B$, which was then found to relax to the desired 1.5 $\mu_B$. A second relaxation was then performed without the density matrix constraint. Note that spin-orbit coupling was not included, meaning that spin-only Pr moments of 2 and 1 $\mu_B$ were obtained in the two states.

In terms of calculation details, F alignment of the Co and Pr moments was initialized in both states. While a full investigation of the stability of this F configuration vs. variables such as strain, $U_{Pr}$, and $U_{Co}$ (the $U$ values on Pr and Co) is beyond the scope of this paper, we performed tests for the $U_{Pr}$ and $U_{Co}$ reported here, which confirm energetic preference for F alignment. The energy difference between F and anti-F configurations ($\sim10^{-3}$ eV/formula unit) is two orders of magnitude smaller than that between metallic and insulating states, however, implying that the magnetic configuration is of little significance for our purposes. All calculations were performed using the PBE exchange-correlation functional revised for solids[77] as implemented in VASP[78], in combination with the Dudarev (effective) Hubbard $U_{eff}$ correction[79] for both Pr $f$ and Co $d$ states. We used the VASP Pr PAW potential with $f$ electrons in the valence states[80]. Results for $U_{Pr} = 4$ eV and $U_{Co} = 3$ eV are shown in Fig. 6b, although, as shown in Fig. S11a-c, the same qualitative trend as in Fig. 6b holds over a range of $U_{Pr}$ and $U_{Co}$. A Monkhorst-Pack grid of $4 \times 4 \times 4$ $k$-points was used to sample reciprocal space for all values of biaxial strain. We used a planewave energy cutoff of 550 eV, and relaxed the internal coordinates, as well as the lattice constant normal to the biaxially-strained plane, until forces converged below $10^{-3}$ eV/Å.

## Data availability

All data needed to evaluate the conclusions in this paper are present in the paper and/or the Supplementary Information. These data have also been deposited in DRUM (the Data Repository for the University of Minnesota (https://conservancy.umn.edu/drum), with the following persistent identifier: https://doi.org/10.13020/7wff-1y61.

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

## Acknowledgements

We gratefully acknowledge helpful comments and input from R. Fernandes. Work at the University of Minnesota (UMN) was primarily funded by the Department of Energy (DOE) through the UMN Center for Quantum Materials under Grant Number DE-SC0016371 (CL). Electron microscopy by SG and KAM was supported by the National Science Foundation (NSF) through the UMN MRSEC under DMR-2011401 (KAM). Parts of the work were performed in the Characterization Facility, UMN, which receives partial support from the NSF through the MRSEC and NNCI programs. Part of this work also used resources at the Spallation Neutron Source, a DOE Office of Science User Facility operated by the Oak Ridge National Laboratory. Aspects of this work additionally used resources of the Advanced Photon Source, a DOE Office of Science User Facility operated by Argonne National Laboratory under Contract No. DE-AC02-06CH11357. Certain commercial equipment, instruments, or materials are identified in this paper to foster understanding. Such identification does not imply recommendation or endorsement by the National Institute of Standards and Technology, nor does it imply that the materials or equipment identified are necessarily the best available for the purpose.

## Author contributions

CL conceived the study (in collaboration with VC) and supervised all aspects of its execution. VC, WMP, JED, CK, AJ, and LF, synthesized and characterized the films (with assistance from JGB); electrical and magnetic measurements were performed by VC, WMP, and JED. STEM/EDX/EELS measurements and analyses were performed by SG and HC with input from KAM. PNR measurements were performed by PQ, PPB, BJK, AH, TC, and MRF and the data were analyzed by PQ, PPB, and VC. SXRD measurements were performed by VC, WMP, and HZ. DFT calculations were performed by DG and TB. CL and VC wrote the paper with input from all authors.

## Competing interests

The authors declare no competing interests.
