## [Peer Review File · Nature Communications]

Room-Temperature Valence Transition in a Strain-Tuned Perovskite OxideThis manuscript does a solid job demonstrating a heteroepitaxial strain induced metal-insulator transition with the concurrent change of magnetism in complex perovskite $(\text{Pr}_{1-y}\text{Y}_y)_{1-x}\text{Ca}_x\text{CoO}_{3-\delta}$ (PYCCO). In particular, the authors describe their attempt to fully control the strain using multiple substrates, and summarize “strain phase diagram” of $(\text{Pr}_{0.85}\text{Y}_{0.15})_{0.7}\text{Ca}_{0.3}\text{CoO}_{3-\delta}$ via structural, electronic transport, polarized neutron reflectometry, and electron energy loss spectra characterizations. They highlighted the enhancement of transition temperature T_{VT} in the nonmagnetic-insulator regime of PYCCO films under a strong compressive strain, especially for the component of $(\text{Pr}_{0.75}\text{Y}_{0.25})_{0.7}\text{Ca}_{0.3}\text{CoO}_{3-\delta}$ which possesses T_{VT} around room temperature. As the similar T_{VT} enhancement has been reported in PYCCO related bulk systems, the originality of this work is to introduce “strain” into PYCCO films. The influence of strain effect, however, has not been adequately discussed, thereby the key word in title “room-temperature valence transition” is inappropriate to cover this work. After careful consideration, we cannot recommend this work to publish in the high impact journal *Nature Communications*.

1. The most critical evidence supporting the title should be the EELS results of $(\text{Pr}_{0.85}\text{Y}_{0.15})_{0.7}\text{Ca}_{0.3}\text{CoO}_{3-\delta}$ exhibited in Fig. 5(b). Here the authors claim that the changes of both the Co ion valence and spin-state are clearly found to be strongly temperature dependent in the sample with -2.10% compressive strain. However, ONLY 0.15 eV difference of the energy on the O K spectra ranging from 153 to 300 K is difficult to clarify their view. In addition, the results of the control sample with +0.25% tensile strain shown in Fig. S7(b), which exhibit similar trend with the case of -2.10% strain, that is, the peak slightly shifts right when temperature increases. Moreover, only the O K results are not enough to discuss the valence transition in Co, a comprehensive comparison among O K, Co $L_{2,3}$ and Pr $M_{4,5}$ spectra is necessary. Thus, the provided results are hard to conclude the valence transition occurring in the PYCCO films.

2. While a nearly room temperature T_{VT} appear in $(\text{Pr}_{0.75}\text{Y}_{0.25})_{0.7}\text{Ca}_{0.3}\text{CoO}_{3-\delta}$, the content in the main text is focused on $(\text{Pr}_{0.85}\text{Y}_{0.15})_{0.7}\text{Ca}_{0.3}\text{CoO}_{3-\delta}$ that exhibit a T_{VT} of 245 K. For the former case, therefore, less characterization were performed, such as the EELS measurement. This is insufficient to support the proposed “room temperature valence transition”. I suggest that the

authors should re-organize the structure of the manuscript, concentrating on evaluating the associated properties of $(\text{Pr}_{0.75}\text{Y}_{0.25})_{0.7}\text{Ca}_{0.3}\text{CoO}_{3-\delta}$.

3. Since the coupled structural/spin-state/metal-insulator transitions have been also introduced in PYCCO related bulk systems, the proposed conclusions of the increase of T_{VT} in PYCCO films induced by strain is not novel, and does not provides substantial advance in the field of complex Cobalt oxides. In my opinion, the strain effect should be seriously investigated for getting insight into the mechanism of the enhanced T_{VT} in PYCCO films beyond the bulk systems.
4. Also, there are several points that should be modified to improve the quality of this manuscript. First, the content arrangement and writing of the manuscript are not well-organized. For instance, since the electronic transport results in Fig. 3 captured an obvious difference of magnetism between compressive/tensile strained PYCCO, the PNR experiments shown in Fig. 4 do not provide further evidence about the magnetism and can be moved to the supplementary. Else, the writing style should be more concise and straightforward, and some typing mistakes, such as “Fig. S6(b, d)” in line 277 must be “Fig. S7(b, d)”, should be corrected.

Reviewer #2:

Remarks to the Author:

Chaturvedi et al present a multi-method study of PrCoO₃-delta where some Pr has been substituted with Ca (dopant) and Y (isovalent). In particular they focus on one composition, referred to as PYCCO, which is a paramagnet in bulk with a valence transition at 135 K from Pr⁴⁺ at low temperature to Pr³⁺ at high temperature. This valence transition is concomitant with an insulator (low temperature) to metal (high temperature) transition and a unit cell expansion without change of symmetry group. Given these interesting phenomena and their interplay, this study - to explore the effects of heteroepitaxial strain - is well-motivated.

The films, which are subjected to a range of strain states both tensile and compressive, appear to be of high quality, as evidenced by the XRD, STEM and AFM data. Magnetometry and transport data reveal some transitions as a function of strain, namely there is a transition in transport if the strain is compressive and a transition in magnetization if the strain is tensile. PNR measurements show that a tensile film has a ferromagnetic ground state while a compressive film shows no long-range order. Meanwhile, STEM-EELS shows that a compressive film undergoes a valence transition while a tensile film does not. These dependencies on strain are consistent with their DFT calculations. I very much like this part of the paper.

The final part of the paper briefly discusses a sample with a greater Y:Pr ratio grown on the most compressive substrate. This sample shows a transition in transport at around room temperature and the only other characterization of this sample is the final figure in the supplementary information with an x-ray diffractogram showing that the unit cell is smaller.

I believe that the authors have done a careful study with mostly robust conclusions. The introduction in particular is very thorough and lays down all the context of the study. This work should be interesting to many people in the community of oxide electronics so I would probably be able to recommend the manuscript for publication in Nature Communications when the following comments are addressed satisfactorily.

1. Given the title, the PYCCO with higher Y:Pr ratio is clearly what the authors want to center as the main selling point of this paper. Given this, I was surprised to see how little this sample is studied. Only one out of seven figures of the main text relate to this film. Why is there no temperature-dependent XRD showing that this sample undergoes the volume change transition at room temperature? Even more importantly, why no characterization of the valence transition directly by EELS? The authors have done a reasonable job of explaining why the transition in transport should be correlated with the valence transition but, to readers, the title and abstract promise more concrete evidence.
2. None of the transport data reported show a metallic phase thus I urge the authors to rethink their usage of the phrase "metal-insulator transition" when it is an insulator-insulator transition.
3. Aside from the transition in transport, the authors claim that the reduced lattice parameter of the film with greater Y:Pr ratio is evidence that the film is already in the "low T" state typical of the material. My question is, what reduction of lattice parameter is anticipated from simply Y having a smaller ionic radius than Pr? Again, temperature-dependent XRD would be valuable here.
4. The PYCCO is expressed with the oxygen content 3-delta. Is oxygen deficiency expected? How is it controlled between samples with different strain? Is it something that is expected to change with temperature?
5. In Figure S7b the oxygen K-edge pre-peak on the LAO substrate is clearly shifting. My estimate is that it shifts approximately 0.2 eV. This is roughly the same as the shift reported for the same sample on YAO yet the authors claim no shift at all. Can they provide the corresponding data for the sample on LAO as Figure 5d? To be clear, I do not think they need the peak shift to claim that there is a transition - the striking change in intensity that is observed on YAO that is not seen on LAO should suffice.
6. On the topic of this pre-peak, is it not possible that they are simply observing a byproduct of the

volume reduction? Meaning, the orbital overlap increases toward low temperature and thus the Co-O hybridization? And this is the cause of the intensity increase rather than a change in valence?

7. If the PYCCO on STO is relaxed, why does it not exhibit the properties of the bulk material? Although little data for that sample is shown, the phase diagram in Figure 6 marks it as having a long-range ordered ferromagnetic ground state.

8. The PNR fits for the film on YAO were done assuming some unwanted overlayer on top of the sample. Why is this not present for the sample on LSAT? How bad was the fit without this overlayer (or with the overlayer on LSAT)?

9. On page 7 it states that Laue fringes are "more abundant under compressive strain". Do they mean "more pronounced" or similar? Because the number of Laue fringes should not change with strain, only thickness.

10. Figure 3 a), the caption mentioned a "thin black line" but the plot shows a thick, dashed teal line.

11. Figure S7c, this is related to my comment on the peak shift, but I find the guide lines a little strange. The dashed line is clearly on the high energy shoulder of the pre-peak at high temperature (yellow) while the solid line is clearly on the low energy side of the lower temperature data (purple). Judging from where they intersect the energy axis, this would indicate a pre-peak shift of at least 0.5 eV, which is in conflict with the reported fits to this peak in the main text. I do not think these lines serve any purpose except an attempt to make the peak shift look larger than it actually is, which, as I mentioned above, is not necessary in my opinion.

12. Although the introduction is nice, it is helpful for readers who are not familiar with the material to be explicit when describing the transitions and state the low and high temperature states. For example, the metal-insulator transition is mentioned but not which state is at high temperature and which at low temperature.

13. The abstract mentioned the possibility of a potential quantum critical point in the system but this is barely discussed again. I also wonder how relevant such a speculation is given the first-order nature of the transition.

Reviewer #3:

Remarks to the Author:

The manuscript of Chaturvedi et al. presents a study on cobalt oxides compounds which display spin-state crossover transition, accompanied by structural and electronic changes as well. The authors fabricate high quality epitaxial films of Pr-based cobalt oxides, namely, $\text{Pr}_{(1-x)}\text{Ca}_x\text{CoO}_3$ and $(\text{Pr}_{(1-y)}\text{Y}_y)_{(1-x)}\text{Ca}_x\text{CoO}_{(3-\delta)}$ and demonstrate control over the phase diagrams of the compounds by applying different degree of epitaxial strain, from tensile to compressive. One highlight of the paper is the successful tuning of the spin-state transition temperature to room-temperature, which is achieved by playing with the chemical composition of the compound and the applied epitaxial strain.

I appreciated the work presented in the manuscript as it presents an original study with useful results, reviving the interest for the cobaltites in their thin film counterpart. The methodology is well described and the main claims are adequately supported by the measurements and data analysis.

However, before publications, I have some comments and questions for the authors.

1) In the Introduction paragraph, at line 60-61, the authors introduced the PCCO compound and their characteristic first order transition. As you mention the case of LaCoO_3 , for which the spin-state transition is consistently accompanied by a change from paramagnetic to diamagnetic behavior, it would be interesting, for the comparison, to mention if in the (bulk) PCCO case the phase transition is also accompanied by a change in the magnetic behavior, that is, what would be the magnetic ground state of bulk PCCO or Y-doped PCCO. Later in the text (line 116-117), a ferromagnetic phase is mentioned for hole-doped PCCO, so I would clarify earlier what is the current understanding of the magnetic phase diagram of such compounds. Also, in analogy to the LaCoO_3 case, can the author mention how does the valence change affect the spin configuration of Co-ions for PCCO?

2) At line 109-110, when referring to study of epitaxial strain on perovskite oxides, I believe it

would be correct to add a reference to the case of the perovskite nickelates, whose strain-driven metal-insulator (MI) transition has been intensively studied in the past years in epitaxial thin films.

3) When discussing the electrical transport of the thin films, the authors identify a metallic and insulating regime for the films. The 'metallic' phase is associated with a higher conductivity regime, whereas the 'insulating' regime occurs after a sudden change decrease of such a conductivity. However, in the 'metallic' regime identified in fig. 3, the resistivity increases as the temperature decreases, which is not a textbook metal-like behavior. Can the author clarify on the nature of the high-conducting phase of these cobaltites? Is similar behaviour consistent with bulk reports and currently understood in terms of electronic transport description? As the manuscript is already quite long a note in SI could be added.

4) In Fig.3 and its caption, the MI transition is identified as a valence transition. Do the authors associate the MIT to the valence transition by analogy to the bulk case? If yes, I would state it explicitly in the text, as at this point of the discussion the valence transition has not been verified yet in the thin films.

5) At line 211-212 the authors mention an anomaly in the R vs T behaviour of the 'metallic' PYCCO films, which is presented in fig. S5. To be honest, I struggle to see such an anomaly in such a figure. The authors should present a closer zoom in the interesting temperature range in order to help the reader to identify the anomaly.

6) Line 276-277. When presenting the EELS measurements at the O-K edge, a control experiment on LAO/PYCCO films is mentioned. However, the temperature range in which such a control experiment is presented does not extend up to the Curie temperature identified in this film, close to 50 K as stated in the main text. Can the authors present the control experiment in the appropriate temperature range? If not, please discuss in more detail what is the purpose of the control experiment.

7) One important conclusion is that compressive strain helps in stabilizing the valence and MI transitions in PCCO epitaxial films. Do the authors believe that such a result support the current understanding of the VT? For example, in the discussion of the results, it would be useful to comment on the relation between the applied strain and the overlap between the Pr and O orbitals, mentioned in the introduction, and draw related conclusions/perspectives.

I would recommend publication in Nat. Comm. after addressing my comments.

**Response to Review Comments, NCOMMS-22-23114,
“Room-temperature valence transition in a strain-tuned perovskite oxide”,
Chaturvedi *et al.***

We would first like to thank the reviewers for their careful, thoughtful, and detailed comments on our work, which we are certain have resulted in a yet stronger revised version of the paper. This version is substantially modified based on the review comments, *including by addition of a substantial amount of requested additional data and analysis*. Below, we provide point-by-point responses to the review comments, along with brief summaries of the ensuing changes in red. In the attached manuscript, the full changes are highlighted in yellow.

Reviewer 1

This manuscript does a solid job demonstrating a heteroepitaxial strain induced metal-insulator transition with the concurrent change of magnetism in complex perovskite $(\text{Pr}_{1-y}\text{Y}_y)_{1-x}\text{Ca}_x\text{CoO}_{3-\delta}$ (PYCCO). In particular, the authors describe their attempt to fully control the strain using multiple substrates, and summarize “strain phase diagram” of $(\text{Pr}_{0.85}\text{Y}_{0.15})_{0.7}\text{Ca}_{0.3}\text{CoO}_{3-\delta}$ via structural, electronic transport, polarized neutron reflectometry, and electron energy loss spectra characterizations. They highlighted the enhancement of transition temperature T_{VT} in the nonmagnetic-insulator regime of PYCCO films under a strong compressive strain, especially for the component of $(\text{Pr}_{0.75}\text{Y}_{0.25})_{0.7}\text{Ca}_{0.3}\text{CoO}_{3-\delta}$ which possesses T_{VT} around room temperature. As the similar T_{VT} enhancement has been reported in PYCCO related bulk systems, the originality of this work is to introduce “strain” into PYCCO films. The influence of strain effect, however, has not been adequately discussed, thereby the key word in title “room-temperature valence transition” is inappropriate to cover this work. After careful consideration, we cannot recommend this work to publish in the high impact journal *Nature Communications*.

We thank the reviewer or reviewers for acknowledging that our study is solid, but we have to respectfully disagree with critical aspects of the above statements. The review states that “*similar T_{vt} enhancement has been reported in PYCCO related bulk systems*”, thus limiting the originality of our work. However, as shown graphically in Fig. 7(a) and in many other works (*e.g.*, refs. 22, 24, 25, 27, 28, and 61 in our manuscript), the T_{vt} in bulk samples of chemically-substituted PCCO barely exceeds 150 K, topping out in extreme cases at ~175 K. At this point, multiple issues are encountered, such as substituent (*e.g.*, Y) solubility limits, unacceptable levels of oxygen deficiency, *etc.* In fact, there has been no indication in the literature that this ceiling can be exceeded. In our work, we reach even *room temperature*, and do so with the very different approach of *heteroepitaxial strain* (rather than chemically-induced strain). We thus disagree that “*similar enhancement has been reported in bulk samples*”. In our work, both the result (a far greater enhancement) and the approach (heteroepitaxy) are very different. We would add that our approach even has the potential to *further* improve T_{vt} , well beyond room temperature, for example through growth on innovative substrates that provide yet larger compressive strain.

We are also confused by the statement that “*The influence of strain effect, however, has not been adequately discussed*”. This is the sole focus of our study. We devoted seven main manuscript figures and 14 supplementary figures to this topic, combining epitaxy, synchrotron X-ray diffraction, grazing-incidence X-ray reflectivity, reciprocal space mapping, rocking curve

analysis, scanning transmission electron microscopy, energy dispersive X-ray spectroscopy, transport, magnetometry, polarized neutron reflectometry, and temperature-dependent X-ray diffraction and electron energy loss spectroscopy. As clearly acknowledged by the other reviewers, the resulting study is comprehensive and focused strongly on strain effects, which we additionally elucidate *via* complementary DFT calculations.

1. The most critical evidence supporting the title should be the EELS results of $(\text{Pr}_{0.85}\text{Y}_{0.15})_{0.7}\text{Ca}_{0.3}\text{CoO}_{3-\delta}$ exhibited in Fig. 5(b). Here the authors claim that the changes of both the Co ion valence and spin-state are clearly found to be strongly temperature dependent in the sample with -2.10% compressive strain. However, ONLY 0.15 eV difference of the energy on the O *K* spectra ranging from 153 to 300 K is difficult to clarify their view. In addition, the results of the control sample with +0.25% tensile strain shown in Fig. S7(b), which exhibit similar trend with the case of -2.10% strain, that is, the peak slightly shifts right when temperature increases.

Moreover, only the O *K* results are not enough to discuss the valence transition in Co, a comprehensive comparison among O *K*, Co *L*_{2,3} and Pr *M*_{4,5} spectra is necessary. Thus, the provided results are hard to conclude the valence transition occurring in the PYCCO films.

We understand the criticism here and have completely overhauled how we analyze and present our EELS data to address this point. If one focuses solely on the shift in the pre-peak position determined from maximum intensity (as we did previously), then the differences between films on YAO and LAO are in fact minimized, and subject to non-negligible noise in the EELS spectra. Instead, we now analyze the data using derivative analysis to more accurately pinpoint the shifts in the peak intensities (from inflection points), as well as using the peak intensity itself. This results in the new Figs. 5(b) and (d) in the main text, as well as the substantially revised panels in Fig. S7, which now compare the films on YAO and LAO *directly and quantitatively*. In particular, Figs. S7(f) and (i) now highlight the tremendous difference between the EELS O *K*-edge pre-peak behavior in these two cases, unambiguously supporting our claims.

With respect to the final statement that Co and Pr spectra should also be recorded, we emphasize two points. First, we are working here with EELS, in a cryo-STEM, where not all edges are as readily accessible to facile study. Co (*L*_{2,3} at 794,779 eV) and Pr (*M*_{4,5} at 951,931 eV) have higher onset energy and lower cross-sections, which result in noisier core-level EELS spectra. Changes in Pr and Co EELS edges would thus be small and difficult to detect and quantify. Second, if we were to switch to XAS, for example, at a synchrotron source, we would incur delays of months to years to obtain such data in the current climate of synchrotron availability in the US. We simply do not see the obvious benefit. We also wish to highlight here the dichotomy between this point of the reviewer and point 3 below. Point 1 essentially raises the concern that the transition seen in our films may not be of the same valence-driven type as in bulk. But point 3 below says the study

is not novel because the phenomena observed are just as in bulk. In our view, the fact that we see a temperature-dependent transition in structure, magnetism, and transport that is very similar to bulk almost guarantees that we have the same valence-driven phenomenon seen in bulk. When one adds the EELS data that we staunchly defend above, and in our revised manuscript, this conclusion is beyond question. Please note here that this does not limit the originality or novelty of our work, as our key advance is to *control* this behavior with epitaxial strain, which we do fully, including promoting it to room temperature

2. While a nearly room temperature TVT appear in $(\text{Pr}_{0.75}\text{Y}_{0.25})_{0.7}\text{Ca}_{0.3}\text{CoO}_{3-\delta}$, the content in the main text is focused on $(\text{Pr}_{0.85}\text{Y}_{0.15})_{0.7}\text{Ca}_{0.3}\text{CoO}_{3-\delta}$ that exhibit a TVT of 245 K. For the former case, therefore, less characterization were performed, such as the EELS measurement. This is insufficient to support the proposed “room temperature valence transition”. I suggest that the authors should re-organize the structure of the manuscript, concentrating on evaluating the associated properties of $(\text{Pr}_{0.75}\text{Y}_{0.25})_{0.7}\text{Ca}_{0.3}\text{CoO}_{3-\delta}$.

The reviewer is correct that much of our manuscript focuses on the $x = 0.30, y = 0.15$ composition. We selected this composition initially as it is perhaps the most extensively studied in bulk, and it thus served as an ideal starting point. After optimizing the thin-film growth (Figs. 1 and 2 of our manuscript), establishing full strain (Fig. 1), observing the control of the ground state and enhancement of T_{vt} (Figs. 3 and 4), confirming the valence transition (Fig. 5), then mapping a full phase diagram (Fig. 6), we then used this knowledge and understanding to further refine T_{vt} (Fig. 7), ultimately hitting room-temperature with $x = 0.30, y = 0.25$ films. Frankly, we see little point in repeating the entire study at $x = 0.30, y = 0.25$ (a massive exercise) just to generate a new version of Fig. 6(a) with shifted features.

Nevertheless, we acknowledge the reviewers point *and have acquired substantial additional data to address it*. First, Fig. 7 has been altered to add new temperature-dependent X-ray diffraction data confirming that in this film also ($x = 0.30, y = 0.25$), the expected volume collapse is observed, indeed centered around room-temperature. This is just as in Fig. 5(c,d) but now at room temperature. Second, we have altered Fig. S14 to present a much fuller characterization of this composition, which now encompasses not only specular high-resolution X-ray diffraction but also new rocking curve analysis, and new atomic force microscopy data. All findings are similar to the $x = 0.30, y = 0.15$ composition, now placing the characterization level of the $x = 0.30, y = 0.25$ films on a similar footing to the $x = 0.30, y = 0.15$ films. Third, we wish to emphasize that we also did all the measurements required to map strain phase diagrams at several other x and y values, which we show immediately below. The main features of the phase behavior *vs.* strain are preserved in all cases, surely demonstrating that x and y change the details of the phase diagrams, but not the overall behavior.

3. Since the coupled structural/spin-state/metal-insulator transitions have been also introduced in PYCCO related bulk systems, the proposed conclusions of the increase of T_{VT} in PYCCO films induced by strain is not novel, and does not provides substantial advance in the field of complex Cobalt oxides. In my opinion, the strain effect should be seriously investigated for getting insight into the mechanism of the enhanced T_{VT} in PYCCO films beyond the bulk systems. This is essentially a repeat of the point we responded to above. Yes, this coupled phase transition is known to occur in bulk, and yes its temperature is enhanced by certain chemical substitutions. But what we show here is *greatly* more efficient improvement in T_{vt} , reaching *room-temperature*, via the completely different approach of *heteroepitaxial strain engineering*. We would further emphasize that heteroepitaxial strain tuning has become a massive sub-field of complex oxide research, and for very good reason. Enhancing and controlling known bulk phenomena in thin films is one central aim of this field, which has generated hundreds of high impact publications and opened up numerous entirely new avenues of research. We thus disagree that the novelty and advance level in our work are limited. Our DFT calculations in support of our experimental

findings add even further weight to this. Finally, we have to point out an essential contradiction between points 1 and 3 in this review. Point 1 essentially suggests that the transition seen in these films may somehow be different to bulk, despite the extensive similarity. But point 3 essentially suggests that this work on films is just studying the same phenomenon as in bulk. It is thus very difficult to understand the reviewer's perspective. We maintain, simply, that the valence-driven structural/spin-state/metal-insulator transition is the same essential phenomenon as seen in bulk, but that in the epitaxial film case we can comprehensively control it. The latter spans enhancing it to room temperature under compression, and suppressing it entirely (replacing it with ferromagnetic metallicity) under tension.

4. Also, there are several points that should be modified to improve the quality of this manuscript. First, the content arrangement and writing of the manuscript are not well-organized. For instance, since the electronic transport results in Fig. 3 captured an obvious difference of magnetism between compressive/tensile strained PYCCO, the PNR experiments shown in Fig. 4 do not provide further evidence about the magnetism and can be moved to the supplementary. Else, the writing style should be more concise and straightforward, and some typing mistakes, such as "Fig. S6(b, d)" in line 277 must be "Fig. S7(b, d)", should be corrected.

The reviewer cites three pieces of evidence for poor organization, which we address in turn. First, the transport results in Fig. 3 do *indirectly* suggest the onset of ferromagnetic behavior under tension, but magnetometry measurements are *undoubtedly essential* to further support this. Moreover, we wish to emphasize that PNR measurements most certainly *do* add something over magnetometry data. Specifically, this is a low- Q neutron scattering technique. When one observes magnetism with this technique, it is thus *proven*, because of the low- Q nature, to be a *long-range-ordered ferromagnetic component*, something that is not possible to prove with magnetometry. In addition, depth-profiling of the magnetization is possible with this technique, establishing in our case that this is *uniform* ferromagnetism, arising in the bulk of the film, not at the interface with the substrate, or at the surface. Also, with this technique the measurements of magnetization are *absolute*, and free of the background subtraction and calibration issues that plague SQUID magnetometry studies of such very thin complex oxides on thick oxide substrates.

We have modified the wording of the manuscript to make these advantages of PNR clearer.

Second, the review criticizes the writing style. We take great pride in the clarity and precision with which our work is presented, however, no specific criticisms are levelled, and this concern was not shared by the other reviewers. Third, the review cites a single typo. **This typo has been corrected** and we thank the reviewer(s) for pointing it out.

Reviewer 2

Chaturvedi et al present a multi-method study of $\text{PrCoO}_{3-\delta}$ where some Pr has been substituted with Ca (dopant) and Y (isovalent). In particular they focus on one composition, referred to as PYCCO, which is a paramagnet in bulk with a valence transition at 135 K from Pr^{4+} at low temperature to Pr^{3+} at high temperature. This valence transition is concomitant with an insulator (low temperature) to metal (high temperature) transition and a unit cell expansion without change of symmetry group. Given these interesting phenomena and their interplay, this study - to explore the effects of heteroepitaxial strain - is well-motivated.

The films, which are subjected to a range of strain states both tensile and compressive, appear to be of high quality, as evidenced by the XRD, STEM and AFM data. Magnetometry and transport data reveal some transitions as a function of strain, namely there is a transition in transport if the strain is compressive and a transition in magnetization if the strain is tensile. PNR measurements show that a tensile film has a ferromagnetic ground state while a compressive film shows no long-range order. Meanwhile, STEM-EELS shows that a compressive film undergoes a valence transition while a tensile film does not. These dependencies on strain are consistent with their DFT calculations. I very much like this part of the paper.

The final part of the paper briefly discusses a sample with a greater Y:Pr ratio grown on the most compressive substrate. This sample shows a transition in transport at around room temperature and the only other characterization of this sample is the final figure in the supplementary information with an x-ray diffractogram showing that the unit cell is smaller.

I believe that the authors have done a careful study with mostly robust conclusions. The introduction in particular is very thorough and lays down all the context of the study. This work should be interesting to many people in the community of oxide electronics so I would probably be able to recommend the manuscript for publication in Nature Communications when the following comments are addressed satisfactorily.

We thank the referee for acknowledging the good motivation for our work, the care with which it was executed, the robustness of the conclusions, and the broad appeal it is likely to have in the oxide community.

1. Given the title, the PYCCO with higher Y:Pr ratio is clearly what the authors want to center as the main selling point of this paper. Given this, I was surprised to see how little this sample is studied. Only one out of seven figures of the main text relate to this film. Why is there no temperature-dependent XRD showing that this sample undergoes the volume change transition at room temperature? Even more importantly, why no characterization of the valence transition directly by EELS? The authors have done a reasonable job of explaining why the transition in transport should be correlated with the valence transition but, to readers, the title and abstract promise more concrete evidence.

We understand the reviewer's point, which overlaps somewhat with point 2 of Reviewer 1. The reviewer is correct that much of our manuscript focuses on the $x = 0.30, y = 0.15$ composition, where the maximum value of T_{vt} we obtain is 245 K. We selected this composition initially as it is perhaps the most extensively studied in bulk, and thus served as an ideal starting point. After optimizing the thin-film growth (Figs. 1 and 2 of our manuscript), establishing full strain (Fig. 1), observing the control of the ground state and enhancement of T_{vt} (Figs. 3 and 4), confirming the valence transition (Fig. 5), then mapping a full phase diagram (Fig. 6) at this composition, we then built on this knowledge and understanding to further refine T_{vt} (Fig. 7), ultimately hitting room-temperature with $x = 0.30, y = 0.25$ films.

While we don't believe it is necessary to repeat our entire study at the $x = 0.30, y = 0.25$ composition, we nevertheless see the reviewers point *and have acquired substantial additional data to address it*. First, as requested, Fig. 7 has been altered to add new temperature-dependent X-ray diffraction data confirming that in this film also, the expected cell volume collapse is observed, indeed centered around room-temperature. Second, also as requested, we have altered

Fig. S14 to present a much fuller characterization of films at this composition, which now encompasses not only specular high-resolution X-ray diffraction but also a new rocking curve, and new atomic force microscopy data (as well as the raw T -dependent XRD data). All findings are similar to the $x = 0.30, y = 0.15$ composition, now placing the characterization level of the $x = 0.30, y = 0.25$ films on a similar footing to the $x = 0.30, y = 0.15$ films. Third, as noted in the response to Reviewer 1, we also did all the measurements required to map strain phase diagrams at several other x and y values, which we show above. The main features are preserved in all cases, surely demonstrating that x and y change the details of the phase diagrams, but not the overall behavior. Fourth, with respect to additional EELS data, we have to say that we feel this is not necessary. With the temperature-dependent transition we see under compression comprehensively established as being of the same type as in bulk (*i.e.*, valence driven), and with the transition we see at 291 K in $\text{YAO}/(\text{Pr}_{0.75}\text{Y}_{0.25})_{0.70}\text{Ca}_{0.30}\text{CoO}_{3-\delta}$ confirmed to be near identical in terms of both transport and now cell volume collapse (new data), we believe it is certain that this is also the same phenomenon. The independence of the overall nature of the strain phase diagrams presented above on x and y further supports this. We hope that these substantial additional data and arguments are convincing on this point. **We have modified the paper during the discussion of Fig. 7 to reinforce the above points.**

2. None of the transport data reported show a metallic phase thus I urge the authors to rethink their usage of the phrase “metal-insulator transition” when it is an insulator-insulator transition. We certainly understand the reviewers point. Completely consistent with the behavior seen in bulk in these systems, in what we term the metallic phase of PYCCO, the resistivity at room temperature is low (typically near $1 \text{ m}\Omega\text{cm}$), the temperature dependence is weak, but the temperature coefficient of resistance is indeed negative. We would like to emphasize, however, that behavior such as that seen in Fig. 3(c) is nevertheless truly metallic using the most rigorous experimental definition of a metal, *i.e.*, that the $T \rightarrow 0$ extrapolation of the conductivity is finite, implying finite density-of-states at the Fermi level, and thus a Fermi surface. This marginal metallic state is as conductive as systems such as this get, due to their low electronic bandwidth. More explicitly, in bulk, systems such as $\text{La}_{1-x}\text{Sr}_x\text{CoO}_3$ attain a clearly metallic state at high x , with positive temperature coefficient of resistance, and relatively low residual resistivities (our record in epitaxial films is near $60 \mu\Omega\text{cm}$). Moving to $\text{Pr}_{1-x}\text{Ca}_x\text{CoO}_3$ substantially lowers the average A-site cation radius, increasing deviations from 180° Co-O-Co bond angle, and decreasing the e_g -derived bandwidth. In this limit, the cobaltites achieve the marginal metallic state described above, but very rarely positive temperature coefficient of resistance, as attested to in refs. 22, 24, and 61 of the manuscript, for example. **To make this point clear, and to make our use of the terms “metal” and “metal-insulator transition” completely unambiguous, we have added a brief discussions of these points in the manuscript.** We thank the reviewer for urging us to eliminate any ambiguity here.

3. Aside from the transition in transport, the authors claim that the reduced lattice parameter of the film with greater Y:Pr ratio is evidence that the film is already in the “low T” state typical of the material. My question is, what reduction of lattice parameter is anticipated from simply Y having a smaller ionic radius than Pr? Again, temperature-dependent XRD would be valuable

here.

Most importantly, as mentioned above, we have now added the temperature-dependent X-ray diffraction data on this composition that the referee requests. Looking at the new Fig. 7 one sees that at 300 K the cell volume collapse is indeed well underway, quantitatively explaining the low *c*-axis lattice parameter from Fig. S14(a). The effect of the Y substitution alone, without the cell volume collapse due to the transition, is far less significant.

4. The PYCCO is expressed with the oxygen content $3-\delta$. Is oxygen deficiency expected? How is it controlled between samples with different strain? Is it something that is expected to change with temperature?

In these cobaltite systems, which have mean formal Co valence midway between $3+$ and $4+$, due to the instability of formally-tetravalent Co in the octahedral coordination in perovskites, finite δ is essentially always the case. In our manuscript, we thus acknowledge the “ δ ” in the formula for our materials, and keep all growth parameters absolutely constant throughout the work. It is always possible that some variations in final δ nevertheless occur *vs.* strain, but we make three important points in this regard. First, we observe metallicity here under *tensile* strain, which is exactly the opposite of the standard, simple expectations based on strain effects on δ , where tension is expected to generally increase oxygen vacancy formation. Second, detailed studies of cobaltites by us and others (*e.g.*, Refs. 51 and 75 and references therein) have established that the tendency to form oxygen vacancies is actually promoted with both signs of strain. The fact that our strain phase diagram reflects very different behavior under tension and compression is thus another very strong indication that the effects are not unduly influenced by variations in δ . Third, the lowest metallic resistivities we obtain in our work (~ 0.5 m Ω cm) compare very favorably to the best reported in the literature for these materials, even after bulk synthesis in high pressure oxygen. Our high-pressure-oxygen sputter deposition approach actually excels in this regard. We have modified the Methods section of our paper to briefly make these points.

5. In Figure S7b the oxygen K-edge pre-peak on the LAO substrate is clearly shifting. My estimate is that it shifts approximately 0.2 eV. This is roughly the same as the shift reported for the same sample on YAO yet the authors claim no shift at all. Can they provide the corresponding data for the sample on LAO as Figure 5d? To be clear, I do not think they need the peak shift to claim that there is a transition – the striking change in intensity that is observed on YAO that is not seen on LAO should suffice.

This point of the reviewer’s is closely related to Point 1 of Reviewer 1. As noted in response to Reviewer 1, we have completely overhauled how we analyze and present our EELS data to address this point. If one focuses solely on the shift in the pre-peak position determined from maximum intensity (as we did previously), then the differences between films on YAO and LAO are in fact minimized, and subject to non-negligible noise in the EELS spectra. Instead, we now analyze the data using derivative analysis to more accurately pinpoint the shifts (from inflection points), as well as using the peak intensity also (which the reviewer correctly highlights is very strong evidence in and of itself). This results in the new Figs. 5(b) and (d), as well as the substantially revised panels in Fig. S7, which now compare the films on YAO and LAO more directly and quantitatively, as requested. In particular, Figs. S7(f) and (i) now highlight the tremendous

difference between the EELS O *K*-edge pre-peak behavior in these two cases, unambiguously supporting our claims.

6. On the topic of this pre-peak, is it not possible that they are simply observing a biproduct of the volume reduction? Meaning, the orbital overlap increases toward low temperature and thus the Co-O hybridization? And this is the cause of the intensity increase rather than a change in valence?

We make two points in response to this interesting question. First, as is often the case with temperature-dependent metal-insulator transitions, the various changes that are coupled together are so intricately intertwined that it can be difficult, perhaps even futile, to try to establish the primary driving forces for specific changes. This is an issue that has been prominent for decades in vanadium oxides, for example. In our case, the structural transition, spin-state transition, metal-insulator transition, and valence transition, are absolutely intertwined and interconnected. It is thus challenging to understand what part of changes in the EELS spectra, for example, are due to valence changes *vs.* volume changes. The field seems certain, however, that a valence change really does take place here. This is evident not only in EELS, XAS, XANES, and EXAFS spectra (on O, Co, and Pr), but also other probes less sensitive to factors such as hybridization. For example, while unpublished at this stage, we have even seen the valence change manifest in the different crystal field excitations expected of Pr³⁺ and Pr⁴⁺ ions, in inelastic neutron spectra.

7. If the PYCCO on STO is relaxed, why does it not exhibit the properties of the bulk material? Although little data for that sample is shown, the phase diagram in Figure 6 marks it as having a long-range ordered ferromagnetic ground state.

We have modified the paper to make clear that what happens at this thickness on STO is *partial strain relaxation, not full strain relaxation*. Much thicker films would eventually fully relax, at which point bulk behavior would be expected. At the thickness studied here (~30 unit cells), the strain relaxation is clearly detectable, but nowhere near complete. In Fig. 6(a), the in-plane strain plotted for the STO case is just the nominal value, as noted in the caption. Long-range ferromagnetic metallicity is indeed observed, which one would expect to die off as strain fully relaxes in far thicker films.

8. The PNR fits for the film on YAO were done assuming some unwanted overlayer on top of the sample. Why is this not present for the sample on LSAT? How bad was the fit without this overlayer (or with the overlayer on LSAT)?

This is correct. Such overlayers are a frequent “pest” in PNR (and some other) measurements at cryogenic temperatures. If conditions inside the cryostat are not ideal, water, for example, can condense on the surface and grow a low-density layer. As shown in Fig. S6, this can be dealt with at the analysis stage. Given the nature of this phenomenon it is present in some cases but not in others. In terms of the magnitude of the effect, the impact on the fits is non-negligible but not massive by any means. As is often the case, in this specific instance we saw a negative scattering length density in the overlayer (see Fig. S6(c)), as well as a background increase, strongly suggestive of a hydrogen-containing material (*e.g.*, ice). If the reviewer is interested, we

can provide refinement results with and without the overlayer for Fig. S6.

9. On page 7 it states that Laue fringes are “more abundant under compressive strain”. Do they mean “more pronounced” or similar? Because the number of Laue fringes should not change with strain, only thickness.

“More pronounced” is a better way to put this, and we have duly changed the manuscript.

10. Figure 3 a), the caption mentioned a “thin black line” but the plot shows a thick, dashed teal line.

We thank the reviewer for pointing out this error, which has now been corrected.

11. Figure S7c, this is related to my comment on the peak shift, but I find the guide lines a little strange. The dashed line is clearly on the high energy shoulder of the pre-peak at high temperature (yellow) while the solid line is clearly on the low energy side of the lower temperature data (purple). Judging from where they intersect the energy axis, this would indicate a pre-peak shift of at least 0.5 eV, which is in conflict with the reported fits to this peak in the main text. I do not think these lines serve any purpose except an attempt to make the peak shift look larger than it actually is, which, as I mentioned above, is not necessary in my opinion.

As noted above, in response to both Reviewers 1 and 2, we have completely overhauled how we perform and present this analysis. We believe that this also addresses this specific criticism.

12. Although the introduction is nice, it is helpful for readers who are not familiar with the material to be explicit when describing the transitions and state the low and high temperature states. For example, the metal-insulator transition is mentioned but not which state is at high temperature and which at low temperature.

This is a very good point, which could avoid confusion. We have thus made alterations to the Introduction to address this.

13. The abstract mentioned the possibility of a potential quantum critical point in the system but this is barely discussed again. I also wonder how relevant such a speculation is given the first-order nature of the transition.

We understand the concern of the reviewer, but make two points in response. First, we do devote a substantial paragraph of discussion of this issue on page 18 of the manuscript (in the conclusions section). Second, we hope to keep this point in the paper as we believe that this is an exciting future direction opened up by this work. To our knowledge, there has been only one other system proposed to study such physics (see Ref. 71 in the manuscript). In our case, however, this can be done *via* strain, a far cleaner tuning parameter than the chemical substitution in Ref. 71, making this future research avenue very attractive. The reviewer has a good point about the first-order nature of the transition, however, which we now note in the relevant discussion on page 18. Please note also that the ferromagnetic to paramagnetic transition under tensile strain in this system is second-order, as typical.

Reviewer 3

The manuscript of Chaturvedi et al. presents a study on cobalt oxides compounds which display spin-state crossover transition, accompanied by structural and electronic changes as well. The authors fabricate high quality epitaxial films of Pr-based cobalt oxides, namely, $\text{Pr}_{(1-x)}\text{Ca}_x\text{CoO}_3$ and $(\text{Pr}_{(1-y)}\text{Y}_y)_{(1-x)}\text{Ca}_x\text{CoO}(3-\delta)$ and demonstrate control over the phase diagrams of the compounds by applying different degree of epitaxial strain, from tensile to compressive. One highlight of the paper is the successful tuning of the spin-state transition temperature to room-temperature, which is achieved by playing with the chemical composition of the compound and the applied epitaxial strain.

I appreciated the work presented in the manuscript as it presents an original study with useful results, reviving the interest for the cobaltites in their thin film counterpart. The methodology is well described and the main claims are adequately supported by the measurements and data analysis.

We thank the reviewer for the positive assessment regarding the originality of our work, and the strength of the support for our claims.

However, before publications, I have some comments and questions for the authors.

1) In the Introduction paragraph, at line 60-61, the authors introduced the PCCO compound and their characteristic first order transition. As you mention the case of LaCoO_3 , for which the spin-state transition is consistently accompanied by a change from paramagnetic to diamagnetic behavior, it would be interesting, for the comparison, to mention if in the (bulk) PCCO case the phase transition is also accompanied by a change in the magnetic behavior, that is, what would be the magnetic ground state of bulk PCCO or Y-doped PCCO.

This is a good point, which we have now clarified in the Introduction of the revised manuscript. The 90-K first-order spin-state transition is indeed accompanied by a sharp decrease in magnetic susceptibility in bulk PCCO and Y-doped PCCO, as one would expect.

Later in the text (line 116-117), a ferromagnetic phase is mentioned for hole-doped PCCO, so I would clarify earlier what is the current understanding of the magnetic phase diagram of such compounds.

This is another good point, which we have also now clarified in the Introduction of the revised manuscript. Ca doping away from the $x = 0.50$ region can indeed stabilize a ferromagnetic metallic ground state in PCCO. Specifically, starting at $x = 0$ in PCO, initial Ca doping first decreases the conventional spin-crossover temperature. At a critical x , this temperature goes to zero and a ferromagnetic metallic state emerges, which is then suppressed around $x = 0.5$ where the first-order spin-state transition occurs.

Also, in analogy to the LaCoO_3 case, can the author mention how does the valence change affect the spin configuration of Co-ions for PCCO?

The understanding from bulk is far from complete on this issue, but several authors have suggested that the valence transition affects primarily the Co^{3+} ions, rather than the Co^{4+} ions, and that a magnetically inhomogeneous ground state thus results. The latter has been documented by various techniques. See refs. 24 and 66 of the manuscript, for example.

2) At line 109-110, when referring to study of epitaxial strain on perovskite oxides, I believe it would be correct to add a reference to the case of the perovskite nickelates, whose strain-driven metal-insulator (MI) transition has been intensively studied in the past years in epitaxial thin films.

We thank the reviewer for this suggestion. We have added citations to a couple of fairly recent reviews on this topic.

3) When discussing the electrical transport of the thin films, the authors identify a metallic and insulating regime for the films. The ‘metallic’ phase is associated with a higher conductivity regime, whereas the ‘insulating’ regime occurs after a sudden change decrease of such a conductivity. However, in the ‘metallic’ regime identified in fig. 3, the resistivity increases as the temperature decreases, which is not a textbook metal-like behavior. Can the author clarify on the nature of the high-conducting phase of these cobaltites? Is similar behaviour consistent with bulk reports and currently understood in terms of electronic transport description? As the manuscript is already quite long a note in SI could be added.

We understand this point, which is very similar to point 2 of Reviewer 2. Please see our response there and the associated changes made to the manuscript.

4) In Fig.3 and its caption, the MI transition is identified as a valence transition. Do the authors associate the MIT to the valence transition by analogy to the bulk case? If yes, I would state it explicitly in the text, as at this point of the discussion the valence transition has not been verified yet in the thin films.

That the MIT in bulk is indeed a valence transition is now accepted, and explicitly discussed in the Introduction. We understand the reviewer’s point here about films, however, and have modified the manuscript at this juncture to make clear that: (a) At this point in the paper we identify the MIT with a valence transition by analogy with bulk, and (b) that we will *directly verify* this point later in the manuscript (*via* the cell volume collapse and EELS data).

5) At line 211-212 the authors mention an anomaly in the R vs T behaviour of the ‘metallic’ PYCCO films, which is presented in fig. S5. To be honest, I struggle to see such an anomaly in such a figure. The authors should present a closer zoom in the interesting temperature range in order to help the reader to identify the anomaly.

To address this point, we now provide in this figure accompanying Zbrodskii plots, where the derivative analysis clearly highlights the anomaly near the Curie temperature.

6) Line 276-277. When presenting the EELS measurements at the O-K edge, a control experiment on LAO/PYCCO films is mentioned. However, the temperature range in which such a control experiment is presented does not extend up to the Curie temperature identified in this film, close to 50 K as stated in the main text. Can the authors present the control experiment in the appropriate temperature range? If not, please discuss in more detail what is the purpose of the control experiment.

We humbly apologize for the confusion here. On that line of the manuscript we incorrectly referred to Fig. S6, when we meant to refer to Fig. S7. The latter figure, the one we meant to refer to, indeed deals with the appropriate temperature range. **This typo has been corrected.**

7) One important conclusion is that compressive strain helps in stabilizing the valence and MI transitions in PCCO epitaxial films. Do the authors believe that such a result support the current understanding of the VT? For example, in the discussion of the results, it would be useful to comment on the relation between the applied strain and the overlap between the Pr and O orbitals, mentioned in the introduction, and draw related conclusions/perspectives.

We do believe that the compressive strain stabilization of the MIT is consistent with the current understanding of the valence transition. We say this because the epitaxial compression favors the *low-cell-volume state*, which is the low- T , low-spin, insulating phase in these systems.

I would recommend publication in Nat. Comm. after addressing my comments.

We hope that the above responses comprehensively address the reviewer's thoughtful points.

Reviewers' Comments:

Reviewer #1:

Remarks to the Author:

First of all, I would like to express my deepest gratitude to the authors for sincerely reply to my previous comments. I understand that the authors have made great effort for making replies.

However, I still don't think that the work satisfies high publication standards from Nature Communications for the two following issues. Firstly, as I and reviewer 2 pointed out, the manuscript mainly focused on the $(\text{Pr}_{0.85}\text{Y}_{0.15})_{0.7}\text{Ca}_{0.3}\text{CoO}_{3-\delta}$ sample with transition temperature of 245 K, while the main selling point of the manuscript is room-temperature valence transition. Secondly, the direct characterization of valence transition in the manuscript was performed by EELS measurements on the O K-edge pre-peak, which I found not conclusive enough. Therefore, the two main selling points epitaxial strain-driven "room-temperature" and "valence transition" are not strongly proved. My recommendation is unchanged, and I cannot recommend this work to publish in the high impact journal Nature Communications.

Reviewer #2:

Remarks to the Author:

The authors have done a commendable job of revising their manuscript. In my opinion it is acceptable as is.

Reviewer #3:

Remarks to the Author:

I read this revised version of the manuscript and I appreciated the work done by the authors. I believe the quality of the manuscript has improved and I recommend publication of the manuscript of Chaturvedi et al. in Nature Communications.

**Response to Second-Round Review Comments, NCOMMS-22-23114A,
“Room-temperature valence transition in a strain-tuned perovskite oxide”,
Chaturvedi *et al.***

We would like to earnestly thank the reviewers for the careful, thoughtful, and detailed comments on our work, which clearly resulted in a strengthened paper. Below, we provide brief responses to the second-round comments, along with brief summaries of the ensuing changes in red. In the attached manuscript, the full changes are highlighted in yellow.

Reviewer 1

First of all, I would like to express my deepest gratitude to the authors for sincerely reply to my previous comments. I understand that the authors have made great effort for making replies.

Thank you. The input of the reviewer is also greatly appreciated.

However, I still don't think that the work satisfies high publication standards from Nature Communications for the two following issues. Firstly, as I and reviewer 2 pointed out, the manuscript mainly focused on the $(\text{Pr}_{0.85}\text{Y}_{0.15})_{0.7}\text{Ca}_{0.3}\text{CoO}_{3-\delta}$ sample with transition temperature of 245 K, while the main selling point of the manuscript is room-temperature valence transition.

We comprehensively addressed this point in the last response, and our comments seem to have convinced Reviewer 2 (see below). We studied the $x = 0.3, y = 0.15$ composition initially as that sort of composition is the best-studied in bulk. Having discovered and fully characterized the strain-tuning capability at this composition, we then switched to $x = 0.3, y = 0.25$ to enable the final promotion of the transition to room temperature with compressive strain. Critically, in response to the last round of comments we added substantial characterization data on the new composition, bringing the characterization level of it to a par with the first composition, as acknowledged by the other reviewers. In essence, our paper now characterizes *both* compositions in detail. We don't really see how there can be any doubt that the essentially identical understanding from the two compositions is sufficient to fully support all of our claims.

Nevertheless, to try to better make our case to readers, we have revised the paper again to make yet clearer why we chose to focus on $x = 0.3, y = 0.15$ initially, and $x = 0.3, y = 0.25$ later.

Secondly, the direct characterization of valence transition in the manuscript was performed by EELS measurements on the O K-edge pre-peak, which I found not conclusive enough. Therefore, the two main selling points epitaxial strain-driven “room-temperature” and “valence transition” are not strongly proved. My recommendation is unchanged, and I cannot recommend this work to publish in the high impact journal Nature Communications.

This point was also addressed as comprehensively as possible in our last response, to, it seems, the satisfaction of the other two reviewers. Our O K-edge EELS data, particularly after our substantial re-analysis, clearly demonstrate a bulk-like coupled spin-state/valence change, which is all we need to claim here to fully make our point. In future work, it would be ideal to add characterization of both the Co and Pr edges in multiple other forms of spectroscopy, but this is surely beyond any

reasonable burden of proof for this discovery. To try to address this point, we have added a couple of lines to the paper to clarify our stance, and invite/direct future work in this direction.

Reviewer 2

The authors have done a commendable job of revising their manuscript. In my opinion it is acceptable as is.

We thank the reviewer for this acknowledgement.

Reviewer 3

I read this revised version of the manuscript and I appreciated the work done by the authors. I believe the quality of the manuscript has improved and I recommend publication of the manuscript of Chaturvedi et al. in Nature Communications.

We thank the reviewer for this acknowledgement.